# PSDNorm: Temporal Normalization for Deep Learning in Sleep Staging

**Théo Gnassounou**
Université Paris-Saclay,
Inria, CEA,
91120 Palaiseau, France
`theo.gnassounou@inria.fr`

**Antoine Collas**
Université Paris-Saclay,
Inria, CEA,
91120 Palaiseau, France
`antoine.collas@inria.fr`

**Rémi Flamary**
École Polytechnique,
IP Paris, CMAP, UMR 7641,
91120 Palaiseau, France
`remi.flamary@polytechnique.edu`

**Alexandre Gramfort**[*]
Université Paris-Saclay,
Inria, CEA,
91120 Palaiseau, France
`alexandre.gramfort@inria.fr`

## ABSTRACT

Distribution shift poses a significant challenge in machine learning, particularly in biomedical applications using data collected across different subjects, institutions, and recording devices, such as sleep data. While existing normalization layers, BatchNorm, LayerNorm and InstanceNorm, help mitigate distribution shifts, when applied over the time dimension they ignore the dependencies and auto-correlation inherent to the vector coefficients they normalize. In this paper, we propose PSDNorm that leverages Monge mapping and temporal context to normalize feature maps in deep learning models for signals. Evaluations with architectures based on U-Net or transformer backbones trained on 10K subjects across 10 datasets, show that PSDNorm achieves state-of-the-art performance on unseen left-out datasets while being more robust to data scarcity.

## 1 INTRODUCTION

**Data Shift in Physiological Signals.** Machine learning techniques have achieved remarkable success in various domains, including computer vision, biology, audio processing, and language understanding. However, these methods face significant challenges when there are distribution shifts between training and evaluation datasets (Moreno-Torres et al., 2012). For example, in biological data, such as electroencephalography (EEG) signals, the distribution of the data can vary significantly. Indeed, data is collected from different subjects, electrode positions, and recording conditions. This paper focuses on sleep staging, a clinical task that consists in classifying periods of sleep in different stages based on EEG signals (Stevens & Clark, 2004). Depending on the dataset, the cohort can be composed of different age groups, sex repartition, health conditions, and recording conditions (O'Reilly et al., 2014; Quan et al., 1998; Marcus et al., 2013). Such variability brings shift in the distribution making it challenging for the model to generalize to unseen datasets.

**Normalization to Address Data Shift.** Normalization layers are widely used in deep learning to improve training stability and generalization. Common layers include BatchNorm (Ioffe & Szegedy, 2015), LayerNorm (Ba et al., 2016), and InstanceNorm (Ulyanov, 2016), which respectively compute statistics across the batch, normalize across all features within each sample, and normalize each channel independently within a sample. Some normalization methods target specific tasks, such as EEG covariance matrices (Kobler et al., 2022) or time-series forecasting (Kim et al., 2021), but they do not fully address spectral distribution shifts reflected in the temporal auto-correlations of signals. Other papers have proposed to adapt layer statistics to new domains (Li et al., 2016; Chang et al., 2019). In sleep staging, a simple normalization is often applied as preprocessing, e.g., standardizing

---

[*]A. Gramfort joined Meta and can be reached at `agramfort@meta.com`

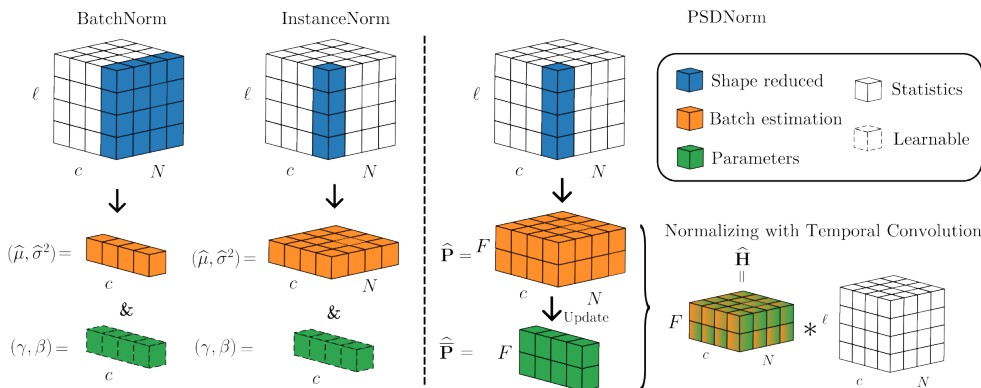

Figure 1: **Description of normalization layers.** The input shape is $(N, c, \ell)$ with batch size $N$, channels $c$, and signal length $\ell$. BatchNorm estimates the mean $\widehat{\mu}$ and variance $\widehat{\sigma}^2$ over batch and time, and learns parameters $(\gamma, \beta)$ to normalize the input. PSDNorm estimates PSDs $\widehat{\mathbf{P}}$ over time and accounts for local temporal correlations. It computes the barycenter PSD $\overline{\widehat{\mathbf{P}}}$, updates it via a running Riemannian barycenter (6), and applies the filter $\widehat{\mathbf{H}}$ to normalize the input. The hyperparameter $f$ controls the extent of temporal correlation considered, thereby adjusting the strength of the normalization.

signals over entire nights (Apicella et al., 2023) or short temporal windows (Chambon et al., 2018). Recent studies (Gnassounou et al., 2023; 2024) highlight the importance of considering temporal correlation and spectral content in normalization, proposing Temporal Monge Alignment (TMA), which aligns Power Spectral Density (PSD) to a common reference using Monge mapping, going beyond simple z-score normalization. However, these methods remain preprocessing steps that cannot be inserted as layers in the network architecture as it is done with BatchNorm, LayerNorm or InstanceNorm.

**Deep Learning for Sleep Staging.** Sleep staging has been addressed by various neural network architectures, which process raw signals (Chambon et al., 2018; Perslev et al., 2021; Guillot & Thorey, 2021), spectrograms (Phan et al., 2023; 2019), or both (Phan et al., 2022a). More recent approaches involve transformer-based models that handle multimodal (Wang et al.), spectrogram (Phan et al., 2022b), or heterogeneous inputs (Guo et al., 2024), offering improved modeling of temporal dependencies. However, most existing models are trained on relatively small cohorts, typically consisting of only a few hundred subjects, which limits their ability to generalize to diverse clinical settings. Notable exceptions include U-Sleep (Perslev et al., 2021), which was trained on a large-scale dataset and incorporates BatchNorm layers to mitigate data variability, and foundational models (Thapa et al.; Fox et al.; Deng et al.) that achieve strong generalization from vast amount of data but require significant computational resources and are challenging to adapt without fine-tuning. Our focus is on developing smaller, efficient models that balance good generalization with ease of training and deployment in clinical practice.

**Contributions.** In this work, we introduce the PSDNorm deep learning layer, a novel approach to address distribution shifts in machine learning for signals. PSDNorm leverages Monge Mapping to incorporate temporal context and normalize feature maps effectively. This layer enhances model robustness to new subjects at inference time. Unlike standard normalization layers such as LayerNorm or InstanceNorm, PSDNorm leverages the sequential nature of intermediate feature maps, as illustrated in Figure 1. We evaluate PSDNorm through extensive experiments on 10 sleep datasets. This evaluation covers 10M of samples across 10K subjects, using a leave-one-dataset-out (LODO) protocol with 3 different random seeds. To the best of our knowledge, such a large-scale and systematic evaluation has never been conducted before. PSDNorm achieves state-of-the-art performance and requires 4 times fewer labeled data to match the accuracy of the best baseline. Results highlight the potential of PSDNorm as a practical and efficient solution for tackling domain shifts in signals.

The paper is structured as follows: Section 2 discusses existing normalization layers and preprocessing. Section 3 introduces PSDNorm, followed by numerical results in Section 4.

**Notations.** Vectors are denoted by small cap boldface letters (e.g., $\mathbf{x}$), matrices by large cap boldface letters (e.g., $\mathbf{X}$). The element-wise product, power of $n$ and division are denoted $\odot$, $\cdot^{\odot n}$ and $\oslash$, respectively. $[\![1, K]\!]$ denotes $\{1, \ldots, K\}$. The absolute value is $|.|$. The discrete circular convolution along the temporal axis operates row-wise as, $* : \mathbb{R}^{c \times \ell} \times \mathbb{R}^{c \times f} \to \mathbb{R}^{c \times \ell}$ for $\ell \geq f$. $\text{vec} : \mathbb{R}^{c \times \ell} \to \mathbb{R}^{c\ell}$ concatenates rows of a time series into a vector. $x_l = [\mathbf{x}]_l$ refers to the $l^{\text{th}}$ element of $\mathbf{x}$, and $X_{l,m} = [\mathbf{X}]_{l,m}$ denotes the element of $\mathbf{X}$ at the $l^{\text{th}}$ row and $m^{\text{th}}$ column. $\mathbf{X}^*$ and $\mathbf{X}^\top$ are the conjugate and the transpose of $\mathbf{X}$, respectively. $\text{diag}$ puts the elements of a vector on the diagonal of a matrix. $\otimes$ is the Kronecker product. $\mathbf{1}_c$ is the vector of ones of size $c$.

## 2 RELATED WORKS

In this section, we first review classical architectures for sleep staging and fundamental concepts of normalization layers. Then, we recall the Temporal Monge Alignment (TMA) method (Gnassounou et al., 2023) that aligns the PSD of signals using optimal transport.

**Deep Learning for Sleep Staging.** Numerous neural network architectures have been proposed for sleep staging, processing data in different formats as introduced in Section 1. Different types of architectures have been explored, such as convolutional neural networks (CNNs) (Chambon et al., 2018), recurrent neural networks (RNNs) (Supratak et al., 2017; Phan et al., 2019), and more recently transformers (Phan et al., 2022b; Wang et al.; Guo et al., 2024), which have shown promise in modeling temporal dependencies in sleep data. While many models are typically evaluated on a limited number of datasets, the work by (Perslev et al., 2021) introduced U-Sleep, a model trained on a large-scale dataset of sleep recordings. Their architecture, based on U-Time (Perslev et al., 2019), incorporates BatchNorm layers to mitigate data variability, and they employ a domain generalization approach: training a single model on a sufficiently diverse set of domains to ensure it generalizes to unseen datasets without additional adaptation. This architecture is composed of encoder-decoder blocks with skip connections, allowing the model to capture both local and global features of the sleep signals effectively. Each encoder and decoder block consists of convolutional layers followed by BatchNorm and non-linear activation functions, enabling the model to learn robust representations of the input data. A more detailed description of U-Time is provided in Section A.3. **Normalization Layers.** Normalization layers enhance training and robustness in deep neural networks. The most common are BatchNorm (Ioffe & Szegedy, 2015), InstanceNorm (Ulyanov, 2016), and LayerNorm (Ba et al., 2016). BatchNorm normalizes feature maps using batch and time statistics, ensuring zero mean and unit variance. The output is adjusted with learnable parameters. InstanceNorm normalizes each channel per sample using its own statistics, independent of the batch (see Fig. 1). Popular in time-series forecasting, it is used in RevIN (Kim et al., 2021), which reverses normalization after decoding. LayerNorm normalizes across all channels and time steps within each sample, with learnable scaling and shifting. While these normalization layers are widely employed, they operate on vectors ignoring statistical dependence and autocorrelation between their coefficients, which are prevalent when operating on time-series. To address this limitation, the Temporal Monge Alignment (TMA) (Gnassounou et al., 2023; 2024) was introduced as a pre-processing step to align temporal correlations by leveraging the Power Spectral density (PSD) of multivariate signals using Monge Optimal Transport mapping.

**Gaussian Periodic Signals.** Consider a multivariate signal $\mathbf{X} \triangleq [\mathbf{x}_1, \ldots, \mathbf{x}_c]^\top \in \mathbb{R}^{c \times \ell}$ of sufficient length. A standard assumption is that this signal follows a centered Gaussian distribution where sensors are uncorrelated and signals are periodic. This periodicity and uncorrelation structure implies that the signal's covariance matrix is block diagonal, with each block having a circulant structure. A fundamental property of symmetric positive definite circulant matrices is their diagonalization (Gray, 2006) with real and positive eigenvalues in the Fourier basis $\mathbf{F}_\ell \in \mathbb{C}^{\ell \times \ell}$ of elements

$$[\mathbf{F}_\ell]_{l,l'} \triangleq \frac{1}{\sqrt{\ell}} \exp\left(-2i\pi \frac{(l-1)(l'-1)}{\ell}\right), \tag{1}$$

where $l, l' \in [\![1, \ell]\!]$. Hence, we have $\text{vec}(\mathbf{X}) \sim \mathcal{N}(\mathbf{0}, \boldsymbol{\Sigma})$ with $\boldsymbol{\Sigma}$ block-diagonal,

$$\boldsymbol{\Sigma} = (\mathbf{I}_c \otimes \mathbf{F}_\ell) \, \text{diag}\left(\text{vec}(\mathbf{P})\right) (\mathbf{I}_c \otimes \mathbf{F}_\ell^*) \ \in \ \mathbb{R}^{c\ell \times c\ell}, \tag{2}$$

where $\mathbf{P} \in \mathbb{R}^{c \times \ell}$ contains positive entries corresponding to the Power Spectral Density of each sensor. In practice, since we only have access to a single realization of the signal, the PSD is estimated with

only $f \ll \ell$ frequencies, *i.e.,* $\mathbf{P} \in \mathbb{R}^{c \times f}$. This amounts to considering the local correlation of the signal and neglecting the long-range correlations.

**Power Spectral Density Estimation.** The Welch estimator (Welch, 1967) computes the PSD of a signal by averaging the squared Fourier transform of overlapping segments of the signal. Hence, the realization of the signal $\mathbf{X} \in \mathbb{R}^{c \times \ell}$ is decimated into overlapping segments $\{\mathbf{X}^{(1)}, \dots, \mathbf{X}^{(L)}\} \subset \mathbb{R}^{c \times f}$ to estimate the PSD. The Welch estimator is defined as

$$\widehat{\mathbf{P}} \triangleq \frac{1}{L} \sum_{l=1}^{L} \left| \left( (\mathbf{1}_c \mathbf{w}^\top) \odot \mathbf{X}^{(l)} \right) \mathbf{F}_f^* \right|^{\odot 2} \in \mathbb{R}^{c \times f} \;, \tag{3}$$

where $\mathbf{w} \in \mathbb{R}^f$ is the window function such that $\|\mathbf{w}\|_2 = 1$.

$f$**-Monge Mapping.** Let $\mathcal{N}(\mathbf{0}, \boldsymbol{\Sigma}^{(s)})$ and $\mathcal{N}(\mathbf{0}, \boldsymbol{\Sigma}^{(t)})$ be source and target centered Gaussian distributions respectively with covariance matrices following the structure (2) and PSDs denoted by $\mathbf{P}^{(s)}$ and $\mathbf{P}^{(t)} \in \mathbb{R}^{c \times f}$. Given a signal $\mathbf{X} \in \mathbb{R}^{c \times \ell}$ such that $\mathrm{vec}(\mathbf{X}) \sim \mathcal{N}(\mathbf{0}, \boldsymbol{\Sigma}^{(s)})$, the $f$-Monge mapping as defined by (Gnassounou et al., 2023; 2024) is

$$m_f\left(\mathbf{X}, \mathbf{P}^{(t)}\right) \triangleq \mathbf{X} * \mathbf{H} \in \mathbb{R}^{c \times \ell}, \quad \text{where} \quad \mathbf{H} \triangleq \frac{1}{\sqrt{f}} \left( \mathbf{P}^{(t)} \oslash \mathbf{P}^{(s)} \right)^{\odot \frac{1}{2}} \mathbf{F}_f^* \in \mathbb{R}^{c \times f} \;. \tag{4}$$

In this case, $f$ controls the alignment between the source and target distributions. Indeed, if $f = \ell$, then the $f$-Monge mapping is the classical Monge mapping between Gaussian distributions and the source signal has its covariance matrix equal to $\boldsymbol{\Sigma}^{(t)}$ after the mapping. If $f = 1$, then each sensor is only multiplied by a scalar.

**Gaussian Wasserstein Barycenter.** For Gaussian distributions admitting the decomposition (2), the Wasserstein barycenter (Agueh & Carlier, 2011) admits an elegant closed-form solution. Consider $K$ centered Gaussian distributions admitting the decomposition (2) of PSDs $\mathbf{P}^{(1)}, \dots, \mathbf{P}^{(K)}$. Their barycenter is also a centered Gaussian distribution $\mathcal{N}(\mathbf{0}, \overline{\boldsymbol{\Sigma}})$ admitting the decomposition (2) with PSD

$$\overline{\mathbf{P}} \triangleq \left( \frac{1}{K} \sum_{k=1}^{K} \mathbf{P}^{(k) \odot \frac{1}{2}} \right)^{\odot 2} \in \mathbb{R}^{c \times f} \;. \tag{5}$$

**Temporal Monge Alignement.** TMA is a pre-processing method that aligns the PSD of multivariate signals using the $f$-Monge mapping. Given a source signal $\mathbf{X}_s$ and a set of target signals $\mathbf{X}_t = \{\mathbf{X}_t^{(1)}, \dots, \mathbf{X}_t^{(K)}\}$, the TMA method uses the $f$-Monge mapping between the source and the Wasserstein barycenter of the target signals. Hence, it simply consists of 1) estimating the PSD of all the signals, 2) computing the Wasserstein barycenter of the target signals, and 3) applying the $f$-Monge mapping to the source signal. TMA, as a preprocessing method, is inherently limited to handling PSD shifts in the raw signals and cannot address more complex distributional changes in the data. This limitation highlights the need for a layer that can effectively capture and adapt to these complex variations during learning and inside deep learning models.

## 3 PSDNorm Layer

The classical normalization layers, such as BatchNorm or InstanceNorm do not take into account the temporal autocorrelation structure of signals. They treat each time sample in the intermediate representations independently. In this section, we introduce the PSDNorm layer that aligns the PSD of each signal onto a barycenter PSD within the architecture of a deep learning model.

PSDNorm is a novel normalization layer that can be used as a drop-in replacement for layers like BatchNorm or InstanceNorm. Instead of simple standardization, it aligns the Power Spectral Density (PSD) of feature maps to a running barycenter PSD. This approach, optimized for modern hardware, enhances model robustness to new subjects at inference time without retraining. We define the normalized feature map as $\widetilde{\mathbf{G}} \triangleq \mathrm{PSDNorm}(\mathbf{G})$. The following sections introduce the core components of PSDNorm and its implementation.

## 3.1 CORE COMPONENTS OF THE LAYER

In the following, we formally define PSDNorm and present each of its three main components: 1) PSD estimation, 2) running Riemannian barycenter estimation, and 3) $f$-Monge mapping computation. Given a batch $\mathcal{B} = \{\mathbf{G}^{(1)}, \ldots, \mathbf{G}^{(N)}\}$ of $N$ pre-normalization feature maps, PSDNorm outputs a normalized batch $\widetilde{\mathcal{B}} = \{\widetilde{\mathbf{G}}^{(1)}, \ldots, \widetilde{\mathbf{G}}^{(N)}\}$ with normalized PSD. Those three steps are detailed in the following and illustrated in the right part of Figure 1.

**PSD Estimation.** First, the estimation of the PSD of each feature map is performed using the Welch method. The per-channel mean $\widehat{\boldsymbol{\mu}}^{(j)}$ is computed for each feature map $\mathbf{G}^{(j)}$ as $\widehat{\boldsymbol{\mu}}^{(j)} \triangleq \frac{1}{\ell} \sum_{l=1}^{\ell} \left[ \mathbf{G}^{(j)} \right]_{:,l} \in \mathbb{R}^c$ .

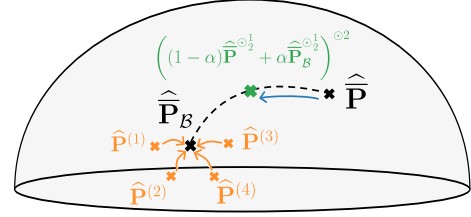

Then, the PSD of the centered feature map $\mathbf{G}^{(j)} - \widehat{\boldsymbol{\mu}}^{(j)} \mathbf{1}_{\ell}^{\top}$, denoted $\widehat{\mathbf{P}}^{(j)}$, is estimated as described in Equation (3). This centering step is required as feature maps are typically non-centered due to activation functions and convolution biases but they are assumed to have a stationary mean. The Welch estimation involves segmenting the centered feature map into overlapping windows, computing the Fourier transform of each window and then averaging them.

Figure 2: **Description of the running Riemanian barycenter.** The barycenter of the batch $\widehat{\overline{\mathbf{P}}}_{\mathcal{B}}$ is estimated from the PSD of each batch sample.

**Geodesic and Running Riemanian Barycenter.**

The PSDNorm aligns the PSD of each feature map to a barycenter PSD. This barycenter is computed during training by interpolating between the batch Wasserstein barycenter and the current running Riemanian barycenter using the geodesic associated with the Bures metric (Bhatia et al., 2019). The batch barycenter is first computed from the current batch PSDs $\{\widehat{\mathbf{P}}^{(1)}, \ldots, \widehat{\mathbf{P}}^{(N)}\}$ using Equation (5). To ensure gradual adaptation, the running barycenter is updated via an exponential geodesic average with $\alpha \in [0, 1]$:

$$\widehat{\overline{\mathbf{P}}} \leftarrow \left( (1-\alpha)\widehat{\overline{\mathbf{P}}}^{\odot\frac{1}{2}} + \alpha\widehat{\overline{\mathbf{P}}}_{\mathcal{B}}^{\odot\frac{1}{2}} \right)^{\odot 2} \in \mathbb{R}^{c \times f} . \tag{6}$$

A proof of the geodesic is provided in Appendix A.1.

**PSD Adaptation with $f$-Monge Mapping.**

The final step of the PSDNorm is the application of the $f$-Monge mapping to each feature map after subtracting the per-channel mean. Indeed, for all $j \in [\![1, N]\!]$, it is defined as

$$\widetilde{\mathbf{G}}^{(j)} = m_f \left( \mathbf{G}^{(j)} - \widehat{\boldsymbol{\mu}}^{(j)} \mathbf{1}_{\ell}^{\top}, \widehat{\overline{\mathbf{P}}} \right) = \left( \left( \mathbf{G}^{(j)} - \widehat{\boldsymbol{\mu}}^{(j)} \mathbf{1}_{\ell}^{\top} \right) * \widehat{\mathbf{H}}^{(j)} \right) \in \mathbb{R}^{c \times \ell} \tag{7}$$

where $\widehat{\mathbf{H}}^{(j)}$ is the Monge mapping filter computed as

$$\widehat{\mathbf{H}}^{(j)} \triangleq \frac{1}{\sqrt{f}} \left( \widehat{\overline{\mathbf{P}}} \oslash \widehat{\mathbf{P}}^{(j)} \right)^{\odot\frac{1}{2}} \mathbf{F}_f^* \in \mathbb{R}^{c \times f} \tag{8}$$

where $\widehat{\mathbf{P}}^{(j)}$ is the estimated PSD of $\mathbf{G}^{(j)} - \widehat{\boldsymbol{\mu}}^{(j)} \mathbf{1}_{\ell}^{\top}$.

## 3.2 IMPLEMENTATION DETAILS

**Overall Algorithm** The forward computation of the proposed layer is outlined in Algorithm 1. At train time, the PSDNorm performs three main operations: 1) PSD estimation, 2) running Riemannian barycenter update, and 3) Monge mapping application. At inference, the PSDNorm operates similarly, except it does not update the running barycenter. The PSDNorm is fully differentiable and can be integrated into any deep learning model. Similarly to classical normalization layers, a stop gradient operation is applied to the running barycenter to prevent the backpropagation of the gradient computation through the barycenter. PSDNorm has a unique additional hyperparameter $f$ which is

the filter size. It controls the alignment between each feature map and the running barycenter PSD and it is typically chosen in our experiments between 1 and 17. In practice, the Fourier transforms are efficiently computed using the Fast Fourier Transform (FFT) algorithm. Because of the estimation of PSDs, the complexity of the PSDNorm, both at train and inference times, is $\mathcal{O}(Nc\ell f \log(f))$, where $N$ is the batch size, $c$ the number of channels, $\ell$ the signal length, and $f$ the filter size.

## 3.3 Discussion and Connections to Related Methods

**PSDNorm as a generalization of InstanceNorm.** InstanceNorm applies a per-channel $z$-score over time, subtracting the mean and dividing by the standard deviation—equivalent to whitening under an i.i.d. assumption over time. In contrast, PSDNorm explicitly accounts for temporal structure by estimating the PSD and whitening/re-coloring in the frequency domain. InstanceNorm is recovered as a special case of PSDNorm by setting the filter size to $f = 1$ and using the uniform PSD barycenter as $\widehat{\overline{\mathbf{P}}} = \mathbf{1}$. as the re-coloring transform instead of the barycentric PSD.

---

**Algorithm 1** Forward pass of PSDNorm

1: **Input:** Batch $\mathcal{B} = \left\{ \mathbf{G}^{(1)}, \ldots, \mathbf{G}^{(N)} \right\}$, running barycenter $\widehat{\overline{\mathbf{P}}}$, filter-size $f$, momentum $\alpha$, training flag
2: **Output:** Normalized batch $\left\{ \widetilde{\mathbf{G}}^{(1)}, \ldots, \widetilde{\mathbf{G}}^{(N)} \right\}$
3: **for** $j = 1$ to $N$ **do**
4: $\quad \widehat{\boldsymbol{\mu}}^{(j)} \leftarrow$ Mean estimation
5: $\quad \widehat{\mathbf{P}}^{(j)} \leftarrow$ PSD est. from $\widetilde{\mathbf{G}}^{(j)} - \widehat{\boldsymbol{\mu}}^{(j)} \mathbf{1}_\ell^\top$ with eq. (3)
6: **end for**
7: **if** training **then**
8: $\quad \widehat{\overline{\mathbf{P}}}_\mathcal{B} \leftarrow$ Batch bary. from $\{\widehat{\mathbf{P}}^{(j)}\}_j$ with eq. (5)
9: $\quad \widehat{\overline{\mathbf{P}}} \leftarrow$ Running bary. up. from $\widehat{\overline{\mathbf{P}}}, \widehat{\overline{\mathbf{P}}}_\mathcal{B}$ with eq. (6)
10: **end if**
11: **for** $j = 1$ to $N$ **do**
12: $\quad \widehat{\mathbf{H}}^{(j)} \leftarrow$ Filter estimation from $\widehat{\mathbf{P}}^{(j)}, \widehat{\overline{\mathbf{P}}}$ with eq. (8)
13: $\quad \widetilde{\mathbf{G}}^{(j)} \leftarrow f$-Monge mapping with eq. (7)
14: **end for**

---

**Similarity with Test-time Domain Adaptation.** PSDNorm is inspired by Temporal Monge Alignment (TMA) (Gnassounou et al., 2023), a pre-processing technique that can be used for test-time adaptation. Test-time Domain Adaptation methods adjust a pre-trained model to a new target domain during inference, without requiring access to the original training data (Wang et al., 2021; Yang et al., 2021). While PSDNorm must be integrated into the model during training and is not a post-hoc adaptation method, it provides a similar benefit at inference time. Designing new modern architectures that incorporate PSDNorm can enhance robustness to domain shifts without the need for retraining or access to source data.

**Discussion of Gaussian and Stationarity Assumptions.** PSDNorm relies on the Gaussian approximation of OT for compensation variability but does not assume that the signals are Gaussian. This allows for efficient alignment of second-order statistics (covariance structure), but also allows preserving higher-order discriminative information. This approach is computationally tractable and targets the most prominent sources of domain shift without over-distorting the signal, a strategy also used in successful methods like Deep CORAL (Sun & Saenko, 2016). Like BatchNorm, PSDNorm assumes shifts are captured by low-order statistics, but it provides a richer alignment by incorporating temporal context.

## 4 Numerical Experiments

In this section, we evaluate the proposed method through a series of experiments designed to highlight its effectiveness and robustness on the clinically relevant task of sleep staging. We first describe the datasets and training setup employed, followed by a performance comparison with existing normalization techniques. Next, we assess the efficiency of PSDNorm by training over varying numbers of subjects per dataset. Finally, we analyze the robustness of PSDNorm against domain shift by focusing on subject-wise performance and different architectures. The code is available at `https://github.com/tgnassou/PSDNorm`. The anonymized code is available in the supplementary material. All numerical experiments were conducted using a total of 1500 GPU hours on NVIDIA H100 GPUs.

## 4.1 EXPERIMENTAL SETUP

**Datasets** To evaluate the effect of normalization layers, we use ten datasets of sleep staging described in Table 1. ABC (Jessie P. et al., 2018), CCSHS (Rosen et al., 2003), CFS (Redline et al., 1995), HPAP (Rosen et al., 2012), MROS (Blackwell et al., 2011), SHHS (Quan et al., 1998), CHAT (Marcus et al., 2013), and SOF (Spira et al., 2008) are publicly available sleep datasets with restricted access from National Sleep Research Resource (NSRR) (Zhang et al., 2018). PHYS (Goldberger et al., 2000) and MASS (O'Reilly et al., 2014) are two other

Table 1: **Characteristics of the datasets.**

| Dataset | Subj. | Rec. | Age ± std | Sex (F/M) |
|---|---|---|---|---|
| ABC | 44 | 117 | 48.8 ± 9.8 | 43%/57% |
| CCSHS | 515 | 515 | 17.7 ± 0.4 | 50%/50% |
| CFS | 681 | 681 | 41.7 ± 20.0 | 55%/45% |
| HPAP | 166 | 166 | 46.5 ± 11.9 | 43%/57% |
| MROS | 2101 | 2698 | 76.4 ± 5.5 | 0%/100% |
| PHYS | 70 | 132 | 58.8 ± 22.0 | 33%/67% |
| SHHS | 5730 | 8271 | 63.1 ± 11.2 | 52%/48% |
| MASS | 61 | 61 | 42.5 ± 18.9 | 55%/45% |
| CHAT | 1230 | 1635 | 6.6 ± 1.4 | 52%/48% |
| SOF | 434 | 434 | 82.8 ± 3.1 | 100%/0% |
| Total | 11032 | 14710 | – | – |

datasets publicly available. Every 30 s epoch is labeled with one of the five sleep stages: Wake, N1, N2, N3, and REM. These datasets are unbalanced in terms of age, sex, number of subjects, and have been recorded with different sensors in different institutions which makes the sleep staging task challenging. We now describe the pre-processing steps and splits of the datasets.

**Data Pre-processing.** We follow a standard pre-processing pipeline used in the field (Chambon et al., 2017; Stephansen et al., 2018). The datasets vary in the number and type of available EEG and electrooculogram (EOG) channels. To ensure consistency, we use two bipolar EEG channels, as some datasets lack additional channels. For dataset from NSRR, we select the channels C3-A2 and C4-A1. For signals from Physionet and MASS, we use the only available channels Fpz-Cz and Pz-Oz. The EEG signals are low-pass filtered with a 30 Hz cutoff frequency and resampled to 100 Hz. All data extraction and pre-processing steps are implemented using MNE-BIDS (Appelhoff et al., 2019) and MNE-Python (Gramfort et al., 2013).

**Leave-One-Dataset-Out (LODO) Setup and Balancing.** We evaluate model performance using a leave-one-dataset-out (LODO) protocol: in each fold, one dataset is held out for testing, and the model is trained on the union of the remaining datasets. From the training data, 80% of subjects are used for training and 20% for validation, which is used for early stopping. The full held-out dataset is used for testing. To assess performance in low-data regimes, we also evaluate a variant in which we subsample at most $N$ subjects per dataset, promoting balanced contributions across training sources. We refer to this configuration as **balanced@$N$**, with $N$ ranging from 40 to 400. The exact number of subjects per dataset in each case is listed in Appendix Table 3.

**Architecture and Training.** Sleep staging has inspired a variety of neural architectures, from early CNN-based models (Chambon et al., 2017; Stephansen et al., 2018; Phan et al., 2022a) to recent attention-based approaches (Phan et al., 2022b; 2023; Wang et al.). We evaluate two architectures: **U-Sleep** (Perslev et al., 2019; 2021), a state-of-the-art temporal CNN model designed for robustness and large-scale training, and a newly introduced architecture, **CNNTransformer**. CNNTransformer combines a lightweight convolutional encoder with a Transformer applied to epoch-level embeddings. It is specifically tailored for two-channel EEG and designed to scale efficiently to large datasets, while remaining minimal in implementation (under 100 lines of code) and training cost (Section A.4). Its design draws inspiration from recent transformer-based models for time series (Yang et al., 2023), with an emphasis on simplicity and practicality.

We use the Adam optimizer (Kingma, 2014) with a learning rate of $10^{-3}$ to minimize the weighted cross-entropy loss, where class weights are computed from the training set distribution. Training is performed with a batch size of 64, and early stopping is applied based on validation loss with a patience of 3 epochs. Each input corresponds to a sequence of 17'30s, with a stride of 10'30s between sequences along the full-night recording. The filter size $f$ of PSDNorm is set to 5. A sensitivity analysis of $f$ is provided in Section A.6 in the appendix, and show that the performance is stable across a range of values from 5 to 11.

**Evaluation.** At inference, the model similarly processes sequences of 17'30s with a stride of 10'30s. Performance is evaluated using the balanced accuracy score (BACC), computed on the central 10'30s of each prediction window. Each experiment is repeated three times with different random seeds, and we report the mean and standard deviation of BACC.

Table 2: **Balanced Accuracy (BACC) scores on the left-out datasets with USleep.** The top section reports results in the **large-scale** setting (using all available subjects), while the bottom section presents results in the **medium-scale** setting (balanced@400). For each row, the best score is highlighted in **bold**, and standard deviations reflect training variability across 3 random seeds. The mean BACC reports the average over all the subjects.

| | Dataset | BatchNorm | LayerNorm | InstanceNorm | TMA | PSDNorm |
|---|---|---|---|---|---|---|
| All subjects | ABC | $78.49_{\pm0.42}$ | $77.94_{\pm0.31}$ | $\mathbf{78.83_{\pm0.59}}$ | $78.33_{\pm0.12}$ | $78.56_{\pm0.67}$ |
| | CCSHS | $\mathbf{88.79_{\pm0.21}}$ | $87.51_{\pm0.77}$ | $88.75_{\pm0.04}$ | $88.61_{\pm0.10}$ | $88.56_{\pm0.36}$ |
| | CFS | $84.97_{\pm0.37}$ | $84.29_{\pm0.67}$ | $\mathbf{85.73_{\pm0.29}}$ | $84.85_{\pm0.13}$ | $85.42_{\pm0.09}$ |
| | CHAT | $64.72_{\pm3.94}$ | $64.36_{\pm0.40}$ | $68.86_{\pm2.49}$ | $69.76_{\pm1.62}$ | $\mathbf{70.57_{\pm1.24}}$ |
| | HOMEPAP | $76.39_{\pm0.29}$ | $75.23_{\pm0.78}$ | $76.70_{\pm0.35}$ | $\mathbf{76.77_{\pm0.66}}$ | $76.72_{\pm0.27}$ |
| | MASS | $73.71_{\pm0.62}$ | $71.39_{\pm3.00}$ | $72.12_{\pm0.70}$ | $\mathbf{73.90_{\pm0.69}}$ | $72.51_{\pm1.68}$ |
| | MROS | $81.30_{\pm0.25}$ | $80.44_{\pm0.29}$ | $81.49_{\pm0.18}$ | $80.91_{\pm0.42}$ | $\mathbf{81.57_{\pm0.34}}$ |
| | PhysioNet | $76.13_{\pm0.57}$ | $75.12_{\pm0.22}$ | $76.15_{\pm0.52}$ | $\mathbf{76.48_{\pm0.37}}$ | $75.96_{\pm1.02}$ |
| | SHHS | $77.97_{\pm1.46}$ | $75.98_{\pm0.48}$ | $79.05_{\pm0.89}$ | $78.21_{\pm0.39}$ | $\mathbf{79.14_{\pm1.01}}$ |
| | SOF | $81.33_{\pm0.54}$ | $81.82_{\pm0.79}$ | $81.98_{\pm0.22}$ | $81.84_{\pm0.49}$ | $\mathbf{82.50_{\pm0.34}}$ |
| | Mean(Dataset) | $78.38_{\pm0.47}$ | $77.41_{\pm0.28}$ | $78.97_{\pm0.11}$ | $78.98_{\pm0.14}$ | $\mathbf{79.15_{\pm0.14}}$ |
| | Mean(Subject) | $78.14_{\pm1.01}$ | $76.78_{\pm0.18}$ | $79.26_{\pm0.48}$ | $78.77_{\pm0.07}$ | $\mathbf{79.51_{\pm0.62}}$ |
| Balanced@400 | ABC | $78.26_{\pm1.33}$ | $75.29_{\pm0.81}$ | $\mathbf{78.73_{\pm0.42}}$ | $78.04_{\pm0.51}$ | $78.18_{\pm0.68}$ |
| | CCSHS | $87.42_{\pm0.16}$ | $85.20_{\pm0.48}$ | $\mathbf{87.62_{\pm0.42}}$ | $87.57_{\pm0.20}$ | $87.58_{\pm0.30}$ |
| | CFS | $84.32_{\pm0.57}$ | $81.66_{\pm1.36}$ | $\mathbf{84.72_{\pm0.33}}$ | $84.58_{\pm0.20}$ | $84.29_{\pm0.36}$ |
| | CHAT | $66.55_{\pm0.88}$ | $61.19_{\pm1.16}$ | $64.43_{\pm4.41}$ | $68.73_{\pm2.48}$ | $\mathbf{70.28_{\pm1.70}}$ |
| | HOMEPAP | $75.25_{\pm0.50}$ | $74.86_{\pm0.25}$ | $76.47_{\pm0.63}$ | $76.10_{\pm0.32}$ | $\mathbf{76.83_{\pm0.61}}$ |
| | MASS | $70.00_{\pm1.91}$ | $68.56_{\pm3.33}$ | $71.52_{\pm1.13}$ | $71.63_{\pm1.92}$ | $\mathbf{72.77_{\pm1.09}}$ |
| | MROS | $\mathbf{80.37_{\pm0.20}}$ | $78.05_{\pm0.22}$ | $80.28_{\pm0.21}$ | $80.09_{\pm0.40}$ | $80.26_{\pm0.11}$ |
| | PhysioNet | $\mathbf{75.81_{\pm0.13}}$ | $71.82_{\pm2.12}$ | $74.68_{\pm0.55}$ | $75.31_{\pm1.54}$ | $74.82_{\pm2.11}$ |
| | SHHS | $76.44_{\pm0.92}$ | $75.12_{\pm0.39}$ | $78.68_{\pm0.37}$ | $77.00_{\pm0.39}$ | $\mathbf{78.88_{\pm0.68}}$ |
| | SOF | $81.08_{\pm1.14}$ | $78.70_{\pm0.50}$ | $80.68_{\pm1.38}$ | $\mathbf{81.25_{\pm0.71}}$ | $79.49_{\pm0.41}$ |
| | Mean(Dataset) | $77.55_{\pm0.34}$ | $75.05_{\pm0.28}$ | $77.78_{\pm0.46}$ | $78.03_{\pm0.35}$ | $\mathbf{78.34_{\pm0.42}}$ |
| | Mean(Subject) | $77.22_{\pm0.34}$ | $75.04_{\pm0.42}$ | $78.17_{\pm0.28}$ | $77.74_{\pm0.36}$ | $\mathbf{78.85_{\pm0.59}}$ |

**Normalization Strategies.** We compare the proposed PSDNorm with three normalization strategies: BatchNorm, LayerNorm, and InstanceNorm. Note that InstanceNorm corresponds to a special case of PSDNorm with $f = 1$ and a fixed identity mapping instead of a learned running barycenter. In the following experiments, the BatchNorm layers in the first three convolutional layers are replaced with either PyTorch's default implementations of LayerNorm, InstanceNorm (Paszke et al., 2019), or PSDNorm. To preserve the receptive field, the filter size $f$ of PSDNorm is used in the first layer and progressively halved in the following ones. We fix the momentum $\alpha$ to $10^{-2}$.

## 4.2 NUMERICAL RESULTS

This section presents results from large-scale sleep stage classification experiments. The analysis begins with a comparison of PSDNorm against standard normalization layers—BatchNorm, LayerNorm, and InstanceNorm—on the full datasets. Then, the data efficiency of each method is evaluated under limited training data regimes. Finally, robustness to distribution shift is assessed via subject-wise performance across multiple neural network architectures.

**Performance Comparison on Full Datasets.** Table 2 (top) reports the LODO BACC of U-Sleep across all datasets, averaged over three random seeds. PSDNorm consistently outperforms all baseline normalization layers—BatchNorm, LayerNorm, InstanceNorm, and TMA—achieving the highest mean BACC of 79.51% over subjects, which exceeds BatchNorm (78.38%), InstanceNorm (78.97%), LayerNorm (77.41%) and TMA (78.77%). On the challenging CHAT dataset, where all methods struggle, PSDNorm outperforms all other normalizations by more than 1 percentage points, highlighting its robustness under strong distribution shifts. Although InstanceNorm is a strong baseline—outperforming BatchNorm and LayerNorm by at least one standard deviation on average—it is consistently surpassed by PSDNorm in average performance. In contrast, LayerNorm underperforms across the board, achieving the lowest average BACC and never ranking first, confirming its limited

suitability for this task. PSDNorm also improves score by almost 1% over TMA, showing that using Monge Alignment inside the network allows for better adaptation.

**Efficiency: Performance with 4× Less Data.**
The PSDNorm layer improves model performance when trained on the full dataset (∼10000 subjects), but such large-scale data availability is not always the case. In many real-world scenarios—such as rare disease studies, pediatric populations, or data collected in constrained clinical settings—labeled recordings are scarce, expensive to annotate, or restricted due to privacy concerns. Evaluating model robustness under these constraints is therefore essential. To this end, we train all models using the balanced@400 setup, which reduces the training data by a factor of 4 compared to the full-data setting. In this lower-data regime, PSDNorm continues to outper-

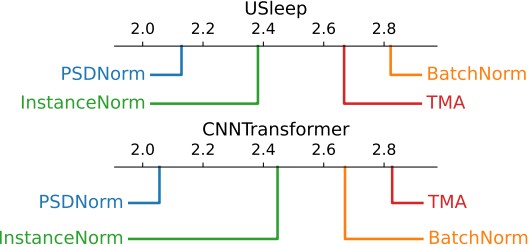

Figure 3: **Critical Difference (CD) diagram for two architectures on datasets balanced @400.** Average ranks across datasets and subjects for USleep and CNNTransformer. Black lines connect methods that are not significantly different.

form all baseline normalization strategies and achieves higher average BACC. The performance improvement of PSDNorm over the best baseline is more pronounced in this setting: the BACC gain reaches +0.67%, compared to +0.25% in the full-data setting. The gains exceed one standard deviation. To assess statistical significance, we conducted a critical difference (CD) test (Demšar, 2006). Figure 3 (top) reports the average rank of each method and the corresponding statistical comparisons. The results confirm that PSDNorm significantly outperforms the baselines, underscoring the value of incorporating temporal structure into normalization for robust and data-efficient generalization. The same trend is observed for U-Sleep trained on all subjects (see in Appendix Figure 8). The following experiments focus on the balanced@400 setup.

**Performance on the most challenging subjects.**
Performance variability across subjects is a key challenge in biomedical applications where ensuring consistently high performance—even for the most challenging subjects—is critical. To highlight the robustness of PSDNorm, Figure 4 presents a scatter plot of subject-wise BACC scores comparing BatchNorm or InstanceNorm vs. PSDNorm across two selected target datasets. CHAT and MASS are two challenging datasets, where the prediction performance is significantly lower than the other datasets. For CHAT, most of the dots are below the diagonal, indicating that PS-DNorm improves performance for 91% of subjects against BatchNorm and 99% of subjects against InstanceNorm, with the largest gains observed for the hardest subjects, reinforcing its ability to han-

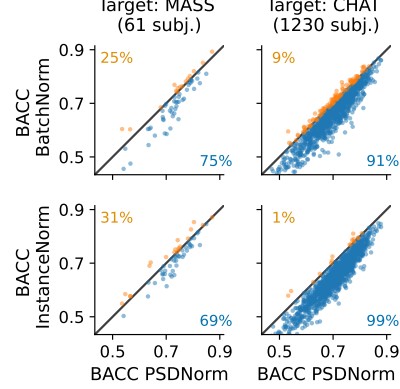

Figure 4: **Subject-wise BACC comparison on MASS and CHAT (balanced @400).** Blue dot means improvement with PSDNorm.

dle challenging cases. For MASS, PSDNorm improves performance for 75% of subjects against BatchNorm and 69% against InstanceNorm. This demonstrates that PSDNorm is not only effective in improving overall performance but also excels in enhancing the performance of the most challenging subjects.

**Robustness Across Architectures.** PSDNorm is a plug-and-play normalization layer that can be seamlessly integrated into various neural network architectures. To demonstrate this flexibility, we evaluate its performance on both the U-Sleep and CNNTransformer models. Figure 3 reports the average rank of each normalization method across datasets and subjects for both architectures using datasets balanced@400. In both architectures, PSDNorm achieves the best overall ranking and demonstrates statistically significant improvements over both BatchNorm, InstanceNorm, and TMA. The results confirm that PSDNorm generalizes well beyond a single architecture and can provide consistent improvements in diverse modeling setups which is not the case of TMA that is ranked the worst with CNNTransformer. InstanceNorm performs competitively in some cases but is never

significantly better than PSDNorm. Detailed numerical scores for CNNTransformer are reported in the supplementary material (Table 7).

It is important to highlight that PSDNorm brings improvements without too much additional computational cost. In appendix Section A.12 we provide a detailed comparison of the computational time of PSDNorm with other normalization layers. The results show that PSDNorm is only slightly slower than BatchNorm and InstanceNorm, with a negligible increase in training time (less than 10%) and no significant impact on inference speed.

### 4.3 Illustration of PSD Normalization

Figure 5 shows how different normalization layers affect the PSD of signals at several stages of the network. The input signals display limited variability, which explains why applying TMA as a pre-processing step provides only marginal benefit. In the first row, corresponding to BatchNorm, the PSD variability increases with depth, a behavior that is undesirable for generalization. TMA exhibits a similar pattern, as no normalization is applied within the network to counteract this accumulation of variance. In contrast, both InstanceNorm and PSDNorm reduce PSD variability across samples. However, InstanceNorm does not fully align the PSDs, and noticeable differences remain between samples. PSDNorm, on the other hand, achieves strong alignment of PSDs across samples, indicating its ability to normalize the underlying temporal correlations. This alignment is essential for improving robustness and generalization, particularly in settings involving distribution shifts.

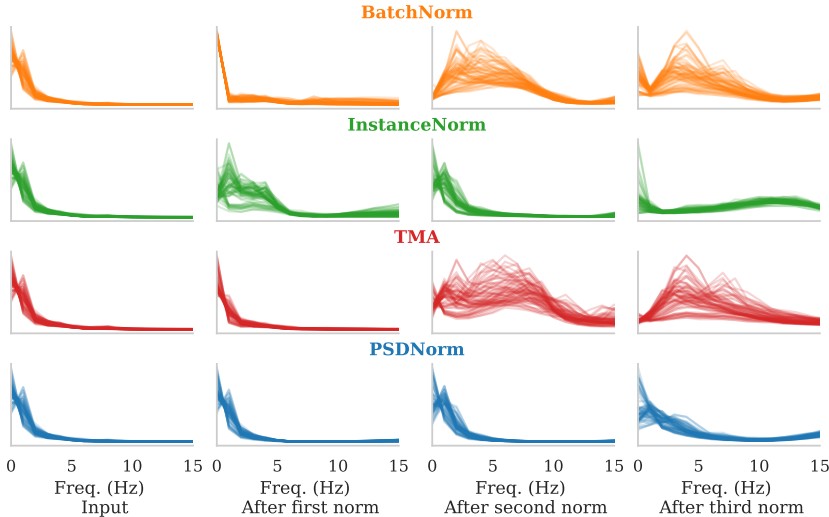

Figure 5: **Illustration of PSD normalization with different normalization layers.** The figure shows the PSD of different segment of 17min from one subject batch as input, and after 3 encoders using different normalization layers.

## 5 Conclusion, Limitations, and Future Work

This paper introduced PSDNorm, a normalization layer that aligns the power spectral density (PSD) of each signal to a geodesic barycenter. By leveraging temporal correlations, PSDNorm offers a principled alternative to standard normalization layers. Experiments on large-scale sleep staging datasets show that PSDNorm consistently improves performance, robustness, and data efficiency, especially under domain shift and limited-data settings—outperforming BatchNorm, LayerNorm, and InstanceNorm across architectures.

While the results are promising, some limitations remain. PSDNorm introduces a filter size hyperparameter ($f$) that controls normalization strength; although we provide default values that perform well across datasets, selecting it automatically in adaptive settings could be challenging.

Despite these limitations, PSDNorm is flexible and easy to integrate into existing models. Future work includes extending it to other signals such as audio and other biomedical applications.

ACKNOWLEDGEMENTS

This work was supported by the grants ANR-22-PESN-0012 to AC under the France 2030 program, ANR-20-CHIA-0016 and ANR-20-IADJ-0002 to AG while at Inria, and ANR-23-ERCC-0006 and ANR-25-PEIA-0005 to RF, all from Agence nationale de la recherche (ANR). This work is supported by Hi! PARIS and ANR/France 2030 program (ANR-23-IACL-0005). This project has also received funding from the European Union's Horizon Europe research and innovation programme under grant agreement 101120237 (ELIAS).

This project received funding from the Fondation de l'École polytechnique

This project was provided with computer and storage resources by GENCI at IDRIS thanks to the grant 2025-AD011016052 and 2025-AD011016067 on the supercomputer Jean Zay's the V100 & H100 partitions.

This work was conducted at Inria, AG is presently employed by Meta Platforms. All the datasets used for this work were accessed and processed on the Inria compute infrastructure.

All the datasets used for this work were accessed and processed on the Inria compute infrastructures. Numerical computation was enabled by the scientific Python ecosystem: Matplotlib Hunter (2007), Scikit-learn Pedregosa et al. (2011), Numpy Harris et al. (2020), Scipy Virtanen et al. (2020), PyTorch Paszke et al. (2019) and MNE Gramfort et al. (2013).

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

# A    APPENDIX

## A.1    PROOF OF THE BURES-WASSERSTEIN GEODESIC (6) BETWEEN COVARIANCE MATRICES OF STRUCTURE (2)

**Proposition A.1.** *Let $\boldsymbol{\Sigma}^{(s)}$ and $\boldsymbol{\Sigma}^{(t)}$ be two covariance matrices in $\mathbb{R}^{cf \times cf}$ following (2). Let us denote $\mathbf{P}^{(s)}$ and $\mathbf{P}^{(t)}$ the corresponding PSD matrices. The geodesic associated with the Bures-Wasserstein metric between $\boldsymbol{\Sigma}^{(s)}$ and $\boldsymbol{\Sigma}^{(t)}$ and parameterized by $\alpha \in [0,1]$ is $\boldsymbol{\Sigma}(\alpha)$ following (2) of PSD*

$$\mathbf{P}(\alpha) = \left( (1-\alpha)\, \mathbf{P}^{(s)\odot\frac{1}{2}} + \alpha \mathbf{P}^{(t)\odot\frac{1}{2}} \right)^{\odot 2}.$$

*Proof.* From Bhatia et al. (2019), the geodesic associated with the Bures-Wasserstein metric between two covariance matrices $\boldsymbol{\Sigma}^{(s)}$ and $\boldsymbol{\Sigma}^{(t)}$ is given by

$$\gamma(\alpha) = (1-\alpha)^2 \boldsymbol{\Sigma}^{(s)} + \alpha^2 \boldsymbol{\Sigma}^{(t)} + \alpha(1-\alpha) \left[ (\boldsymbol{\Sigma}^{(s)}\boldsymbol{\Sigma}^{(t)})^{\frac{1}{2}} + (\boldsymbol{\Sigma}^{(t)}\boldsymbol{\Sigma}^{(s)})^{\frac{1}{2}} \right]. \tag{9}$$

where

$$(\boldsymbol{\Sigma}^{(s)}\boldsymbol{\Sigma}^{(t)})^{\frac{1}{2}} = \boldsymbol{\Sigma}^{(s)\frac{1}{2}} \left( \boldsymbol{\Sigma}^{(s)\frac{1}{2}} \boldsymbol{\Sigma}^{(t)} \boldsymbol{\Sigma}^{(s)\frac{1}{2}} \right)^{\frac{1}{2}} \boldsymbol{\Sigma}^{(s)-\frac{1}{2}}. \tag{10}$$

Since $\boldsymbol{\Sigma}^{(s)}$ and $\boldsymbol{\Sigma}^{(t)}$ diagonalize in the unitary basis $\mathbf{I}_c \otimes \mathbf{F}_f$, $\gamma(\alpha)$ also diagonalizes in this basis. Thus, we only have to compute the geodesic between the PSD matrices $\mathbf{P}^{(s)}$ and $\mathbf{P}^{(t)}$ and from now on, all operations are element-wise. Let $\mathbf{P}(\alpha)$ be the PSD of $\gamma(\alpha)$, we have

$$\mathbf{P}(\alpha) = (1-\alpha)^2\mathbf{P}^{(s)} + \alpha^2\mathbf{P}^{(t)} + \alpha(1-\alpha)\left[ (\mathbf{P}^{(s)} \odot \mathbf{P}^{(t)})^{\odot\frac{1}{2}} + (\mathbf{P}^{(t)} \odot \mathbf{P}^{(s)})^{\odot\frac{1}{2}} \right] \tag{11}$$

$$= (1-\alpha)^2\mathbf{P}^{(s)} + \alpha^2\mathbf{P}^{(t)} + 2\alpha(1-\alpha)(\mathbf{P}^{(s)} \odot \mathbf{P}^{(t)})^{\odot\frac{1}{2}} \tag{12}$$

$$= \left( (1-\alpha)\mathbf{P}^{(s)\odot\frac{1}{2}} + \alpha\mathbf{P}^{(t)\odot\frac{1}{2}} \right)^{\odot 2}. \tag{13}$$

This concludes the proof.    ∎

## A.2    BALANCED DATASETS

Table 3: Number of samples in the balanced datasets. Average and standard deviation (across LODO) are computed over 10 datasets left-out from the training set.

| Balanced datasets | Number of subjects |
|---|---|
| Balanced@40 | $360 \pm 0$ |
| Balanced@100 | $787 \pm 19$ |
| Balanced@200 | $1387 \pm 63$ |
| Balanced@400 | $2466 \pm 157$ |
| All subjects | $9929 \pm 1659$ |

In the main paper, we report results across different training set sizes. Since the datasets are highly imbalanced (*e.g.,* ABC has 44 subjects, SHHS has 5,730), we create balanced subsets by randomly selecting up to $N$ subjects per dataset. This avoids over-representing the largest dataset and ensures greater diversity in the training data. We consider four values of $N$: 40, 100, 200, and 400. The average number of subjects in each balanced set is shown in Table 3. Notably, the balanced set with 400 subjects contains roughly four times less data than the full dataset.

## A.3    U-TIME: CNN FOR TIME SERIES SEGMENTATION

U-Time Perslev et al. (2019; 2021) is a convolutional neural network (CNN) inspired by the U-Net architecture Ronneberger et al. (2015), designed for segmenting temporal sequences. U-Time maps

sequential inputs of arbitrary length to sequences of class labels on a freely chosen temporal scale. The architecture is composed of several encoder and decoder blocks, with skip connections between them.

**Encoder blocks**  A single encoder block is composed of a convolutional layer, an activation function, a BatchNorm layer, and a max pooling layer. First, the convolution is applied to the input signal, followed by the activation function and the BatchNorm layer. Finally, the max pooling layer downsamples the temporal dimension. In the following, the pre-BatchNorm feature map is denoted $\mathbf{G}$ and the post-BatchNorm feature map $\widetilde{\mathbf{G}}$, *i.e.,* $\widetilde{\mathbf{G}} \triangleq \text{BatchNorm}(\mathbf{G})$. Each encoder block downsamples by 2 the signal length but increases the number of channels.

**Decoder blocks and Segmentation Head**  The decoding part of U-Time is symmetrical to the encoding part. Each decoder block doubles the signal length and decreases the number of channels. It is composed of a convolutional layer, an activation function, a BatchNorm layer, an upsampling layer and a concatenation layer of the skip connection of the corresponding encoding block. Finally, the segmentation head applies two convolutional layers with an activation function in between to output the final segmentation. It should be noted that U-Time employs BatchNorm layers but other normalization layers, such as LayerNorm Ba et al. (2016) or InstanceNorm Ulyanov (2016) are possible.

**Implementation**  The architecture is inspired from Braindecode Schirrmeister et al. (2017). The implementation is improved to make it more efficient and faster. One epoch of training takes about 30 min on a single H100 GPU.

## A.4 ARCHITECTURE: CNNTRANSFORMER

The CNNTransformer is a hybrid architecture designed for multichannel time series classification inspired by transformers for EEG-Data Wang et al.; Phan et al. (2022b); Yang et al. (2023); Thapa et al.. It combines convolutional feature extraction with long-range temporal modeling via a Transformer encoder at epoch-level. The model processes an input tensor of shape $(B, S, C, T)$, where $B$ is the batch size, $S$ is the number of temporal segments, $C$ is the number of input channels, and $T$ is the number of time samples per segment. It outputs a tensor of shape $(B, n_{\text{classes}}, S)$, where $n_{\text{classes}}$ is the number of classes and $S$ is the number of epochs.

The architecture consists of the following components:

- **Reshaping:** The input is first permuted and reshaped to a 3D tensor of shape $(B, C, S \cdot T)$ to be compatible with 1D convolutional layers applied along the temporal dimension.
- **CNN-based Feature Extractor:** A stack of 10 Conv1D layers, each followed by ELU activation and Batch Normalization. Some layers use a stride greater than 1 to progressively reduce the temporal resolution. This block extracts local temporal patterns and increases the representational capacity up to a dimensionality of $d_{\text{model}}$.
- **Adaptive Pooling:** An AdaptiveAvgPool1D layer reduces the temporal length to a fixed number of steps ($S$), independent of the input sequence length. This step ensures a consistent temporal resolution before the Transformer.
- **Positional Encoding:** Learnable positional embeddings of shape $(1, S, d_{\text{model}})$ are added to the feature representations to preserve temporal ordering before passing through the Transformer encoder.
- **Transformer Encoder:** A standard Transformer encoder composed of $L$ layers, each consisting of multi-head self-attention and a feedforward sublayer. This module models global temporal dependencies across the $S$ steps.
- **Classification Head:** After transposing the data to shape $(B, d_{\text{model}}, S)$, a final 1D convolution with a kernel size of 1 projects the output to $n_{\text{classes}}$, yielding predictions for each epoch segment.

The model is trained end-to-end using standard optimization techniques. The use of adaptive pooling and self-attention enables it to generalize across variable-length inputs while maintaining temporal resolution. A full summary of the architecture is provided in Table 4.

Table 4: Architecture overview of the CNNTransformer model. In pratice, $d_{\text{model}}$ is set to 768, $n_{\text{head}}$ to 8, and $S$ is 35.

| Stage | Operation | Details | Output Shape |
|---|---|---|---|
| Input | Raw signal | Multichannel EEG signal with $S$ segments and $T$ time samples per segment | $(B, S, C, T)$ |
| Reshape | Permute & flatten | Rearranged as $(B, C, S \cdot T)$ to process with 1D convolutions | $(B, C, S \cdot T)$ |
| Feature Extractor | 1D CNN stack | 10-layer sequence of Conv1D $\rightarrow$ ELU $\rightarrow$ Batch-Norm; includes temporal downsampling via stride | $(B, d_{\text{model}}, T')$ |
| Temporal Pooling | AdaptiveAvgPool1D | Downsamples to fixed temporal resolution defined by $S$ | $(B, d_{\text{model}}, S)$ |
| Positional Encoding | Learnable embeddings | Added to temporal dimension to encode temporal order before transformer layers | $(B, d_{\text{model}}, S)$ |
| Transformer Encoder | Multi-head attention | 2 Transformer layers with $d_{\text{model}}$ embedding dimension, $n_{\text{head}}$ heads, and feedforward sublayers | $(B, d_{\text{model}}, S)$ |
| Classifier | Linear projection | Projects feature vectors to class logits at each epoch time step | $(B, n_{\text{classes}}, S)$ |

## A.5 Equation for BatchNorm and InstanceNorm

**BatchNorm** The BatchNorm layer Ioffe & Szegedy (2015) normalizes features maps in a neural network to have zero mean and unit variance. At train time, given a batch $\mathcal{B} = \{\mathbf{G}^{(1)}, \ldots, \mathbf{G}^{(N)}\} \subset \mathbb{R}^{c \times \ell}$ of $N$ pre-BatchNorm feature maps and for all $j, m, l \in [\![1, N]\!] \times [\![1, c]\!] \times [\![1, \ell]\!]$, the BatchNorm layer is computed as

$$\widetilde{G}_{m,l}^{(j)} = \gamma_m \frac{G_{m,l}^{(j)} - \widehat{\mu}_m}{\sqrt{\widehat{\sigma}_m^2 + \varepsilon}} + \beta_m \ , \tag{14}$$

where $\boldsymbol{\gamma}, \boldsymbol{\beta} \in \mathbb{R}^c$ are learnable parameters. The mean and standard deviation $\widehat{\boldsymbol{\mu}} \in \mathbb{R}^c$ and $\widehat{\boldsymbol{\sigma}} \in \mathbb{R}^c$ are computed across the time and the batch,

$$\widehat{\mu}_m \triangleq \frac{1}{N\ell} \sum_{j=1}^{N} \sum_{l=1}^{\ell} G_{m,l}^{(j)},$$

$$\widehat{\sigma}_m^2 \triangleq \frac{1}{N\ell} \sum_{j=1}^{N} \sum_{l=1}^{\ell} \left( G_{m,l}^{(j)} - \widehat{\mu}_m \right)^2 . \tag{15}$$

At test time, the mean and variance $\widehat{\boldsymbol{\mu}}$ and $\widehat{\boldsymbol{\sigma}}$ are replaced by their running mean and variance, also called exponential moving average, estimated during training.

**InstanceNorm** Another popular normalization is the InstanceNorm layer Ulyanov (2016). During training, InstanceNorm operates similarly to (14), but the mean and variance are computed per sample instead of across the batch dimension, *i.e.,* $\widehat{\mu}_m^{(j)}$ and $\widehat{\sigma}_m^{(j)}$ are computed for each sample $j$,

$$\widehat{\mu}_m^{(j)} \triangleq \frac{1}{\ell} \sum_{l=1}^{\ell} G_{m,l}^{(j)} \ ,$$

$$(\widehat{\sigma}_m^{(j)})^2 \triangleq \frac{1}{\ell} \sum_{l=1}^{\ell} \left( G_{m,l}^{(j)} - \widehat{\mu}_m^{(j)} \right)^2 . \tag{16}$$

Hence, each sensor of each sample is normalized independently of the others. At test time, InstanceNorm behaves identically to its training phase and therefore does not rely on running statistics contrary to the BatchNorm.

## A.6 Sensitivity to Filter Size

The filter size $f$ in PSDNorm controls the temporal context used for normalization, influencing the strength of adaptation to temporal variations. Figure 6 shows the impact of different $f$ values on the

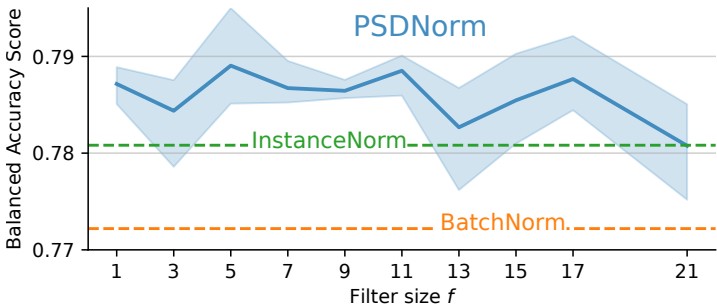

Figure 6: **Performance of PSDNorm with varying filter sizes.** The BACC score is plotted against the filter size used with U-Sleep.

BACC score across datasets using U-Sleep trained on balanced@400. This experiments shows that for any $f$, PSDNorm consistently improves performance compared to other normalization techniques. Taking a $f$ between 5 and 11 yields the best results, with a peak at $f = 5$. Smaller values (e.g., $f = 1$, equivalent to InstanceNorm) provide less adaptation, while larger values (e.g., $f = 21$) may over-smooth temporal variations, leading to diminished performance. Overall, the results shows that $f$ is not so sensitive yielding good performance for a wide range of values.

## A.7  F1 SCORE VS. BALANCED ACCURACY

In the main paper, we report Balanced Accuracy scores, which account for class imbalance in sleep stage classification. Prior work, such as the U-Time paper Perslev et al. (2019), uses the F1 score to evaluate performance. In Table 5, we report F1 scores on the left-out datasets. These scores are slightly higher than the Balanced Accuracy scores and are comparable to those reported in the U-Time paper.

Our main findings remain consistent: BatchNorm and InstanceNorm are the strongest baselines and achieve the best performance on 3 out of 10 datasets. PSDNorm outperforms all other methods on 7 out of 10 datasets. The same trend holds for the balanced@400 setup, where PSDNorm again outperforms all baselines on 7 datasets, while InstanceNorm is never the top performer.

These results confirm that our implementation achieves state-of-the-art performance in sleep stage classification. Moreover, PSDNorm maintains its advantage even in data-limited settings

## A.8  IMPACT OF WHITENING AND TARGET COVARIANCE

As explained in the main paper, InstanceNorm is a special case of PSDNorm with $F = 1$ and an identity target covariance matrix (i.e., whitening). PSDNorm extends this by (i) using temporal context with $F > 1$, and (ii) mapping the PSD to a target covariance matrix, such as a barycenter (i.e., colorization).

In this section, we evaluate the impact of whitening on the performance of PSDNorm, to assess the benefit of using a barycenter as the target covariance matrix. Table 6 reports results on 10 datasets (balanced@400), with and without whitening.

Whitening improves performance on only one dataset (CCSHS), while projecting to the barycenter yields the best results on 6 datasets.

This suggests that, while whitening may help when $F = 1$, it is less effective when $F > 1$. Using a barycenter leads to a more robust and stable target covariance matrix.

## A.9  GENERALIZATION OF PSDNORM IN CNNTRANSFORMER

The CNNTransformer architecture is a hybrid model that combines convolutional and transformer layers for time series classification.

Table 5: **F1 scores of different methods on the left-out datasets.** The lower section displays results for training over datasets balanced @400 *i.e.,* **small-scale dataset**, while the upper section presents results for training over all subjects *i.e.,* **large-scale dataset**. The best scores are highlighted in **bold**. The reported standard deviations indicate performance variability across 3 seeds.

| | Dataset | BatchNorm | LayerNorm | InstanceNorm | TMA | PSDNorm(F=5) |
|---|---|---|---|---|---|---|
| All subjects | ABC | $81.00_{\pm0.11}$ | $79.50_{\pm0.49}$ | $80.56_{\pm0.39}$ | $80.89_{\pm0.06}$ | $\mathbf{81.12_{\pm0.37}}$ |
| | CCSHS | $\mathbf{89.83_{\pm0.19}}$ | $89.01_{\pm0.43}$ | $89.39_{\pm0.16}$ | $89.37_{\pm0.11}$ | $89.13_{\pm0.17}$ |
| | CFS | $88.30_{\pm0.52}$ | $87.39_{\pm0.06}$ | $88.45_{\pm0.17}$ | $88.28_{\pm0.37}$ | $\mathbf{88.52_{\pm0.15}}$ |
| | CHAT | $65.77_{\pm4.06}$ | $65.25_{\pm3.96}$ | $71.35_{\pm2.75}$ | $71.80_{\pm2.66}$ | $\mathbf{72.16_{\pm2.21}}$ |
| | HOMEPAP | $77.06_{\pm0.14}$ | $76.62_{\pm1.06}$ | $77.50_{\pm0.46}$ | $\mathbf{77.82_{\pm0.64}}$ | $77.30_{\pm0.24}$ |
| | MASS | $77.27_{\pm1.42}$ | $74.21_{\pm2.05}$ | $75.12_{\pm2.08}$ | $\mathbf{77.74_{\pm1.05}}$ | $76.00_{\pm3.00}$ |
| | MROS | $\mathbf{85.53_{\pm0.48}}$ | $84.02_{\pm0.95}$ | $85.22_{\pm0.19}$ | $85.13_{\pm0.98}$ | $85.02_{\pm0.42}$ |
| | PhysioNet | $74.98_{\pm1.84}$ | $74.29_{\pm1.50}$ | $75.07_{\pm1.05}$ | $\mathbf{76.01_{\pm0.73}}$ | $75.29_{\pm1.21}$ |
| | SHHS | $78.95_{\pm0.92}$ | $78.04_{\pm1.21}$ | $80.30_{\pm1.29}$ | $78.84_{\pm0.43}$ | $\mathbf{80.32_{\pm0.91}}$ |
| | SOF | $86.30_{\pm0.40}$ | $85.82_{\pm0.22}$ | $86.57_{\pm0.60}$ | $86.31_{\pm0.27}$ | $\mathbf{86.99_{\pm0.33}}$ |
| | Mean(Dataset) | $80.50_{\pm0.51}$ | $79.41_{\pm0.73}$ | $80.95_{\pm0.36}$ | $\mathbf{81.22_{\pm0.20}}$ | $81.19_{\pm0.11}$ |
| | Mean(Subject) | $80.05_{\pm0.78}$ | $79.09_{\pm0.90}$ | $81.31_{\pm0.83}$ | $80.59_{\pm0.19}$ | $\mathbf{81.39_{\pm0.69}}$ |
| Balanced @400 | ABC | $\mathbf{79.80_{\pm0.34}}$ | $77.86_{\pm0.80}$ | $78.36_{\pm1.20}$ | $79.49_{\pm0.68}$ | $78.08_{\pm0.78}$ |
| | CCSHS | $88.32_{\pm0.49}$ | $87.22_{\pm0.51}$ | $88.73_{\pm0.52}$ | $88.47_{\pm0.62}$ | $\mathbf{88.79_{\pm0.99}}$ |
| | CFS | $87.01_{\pm0.18}$ | $85.61_{\pm0.16}$ | $\mathbf{87.62_{\pm0.27}}$ | $87.37_{\pm0.44}$ | $87.06_{\pm0.77}$ |
| | CHAT | $66.56_{\pm1.42}$ | $61.32_{\pm2.25}$ | $64.19_{\pm4.63}$ | $69.90_{\pm2.74}$ | $\mathbf{71.86_{\pm0.95}}$ |
| | HOMEPAP | $76.20_{\pm1.25}$ | $76.15_{\pm1.13}$ | $77.66_{\pm0.58}$ | $76.83_{\pm0.97}$ | $\mathbf{77.85_{\pm1.29}}$ |
| | MASS | $76.06_{\pm1.69}$ | $73.95_{\pm5.80}$ | $76.94_{\pm1.12}$ | $76.32_{\pm0.36}$ | $\mathbf{77.16_{\pm1.73}}$ |
| | MROS | $83.69_{\pm0.39}$ | $82.22_{\pm1.27}$ | $83.95_{\pm0.53}$ | $\mathbf{84.15_{\pm0.46}}$ | $83.51_{\pm0.84}$ |
| | PhysioNet | $\mathbf{76.26_{\pm1.27}}$ | $70.40_{\pm0.14}$ | $73.84_{\pm0.93}$ | $75.24_{\pm2.72}$ | $73.51_{\pm3.05}$ |
| | SHHS | $76.98_{\pm0.70}$ | $75.98_{\pm0.22}$ | $79.12_{\pm0.96}$ | $78.19_{\pm0.90}$ | $\mathbf{79.26_{\pm1.35}}$ |
| | SOF | $85.49_{\pm0.58}$ | $84.23_{\pm1.30}$ | $85.50_{\pm0.86}$ | $\mathbf{85.56_{\pm0.90}}$ | $84.14_{\pm1.05}$ |
| | Mean(Dataset) | $79.64_{\pm0.41}$ | $77.57_{\pm0.73}$ | $79.59_{\pm0.25}$ | $\mathbf{80.15_{\pm0.26}}$ | $80.12_{\pm0.57}$ |
| | Mean(Subject) | $78.57_{\pm0.55}$ | $76.86_{\pm0.22}$ | $79.53_{\pm0.30}$ | $79.70_{\pm0.66}$ | $\mathbf{80.29_{\pm0.68}}$ |

Table 6: Impact of the whitening on the performance of PSDNorm on the 10 datasets balanced @ 400.

| Dataset | BatchNorm | InstanceNorm | PSDNorm | |
|---|---|---|---|---|
| | | | Barycenter | Whitening |
| ABC | $78.26_{\pm1.33}$ | $\mathbf{78.73_{\pm0.42}}$ | $78.18_{\pm0.68}$ | $77.86_{\pm1.33}$ |
| CCSHS | $87.42_{\pm0.16}$ | $87.62_{\pm0.42}$ | $87.58_{\pm0.30}$ | $\mathbf{87.80_{\pm0.23}}$ |
| CFS | $84.32_{\pm0.57}$ | $\mathbf{84.72_{\pm0.33}}$ | $84.29_{\pm0.36}$ | $84.01_{\pm0.60}$ |
| CHAT | $66.55_{\pm0.88}$ | $64.43_{\pm4.41}$ | $\mathbf{70.28_{\pm1.70}}$ | $69.07_{\pm3.73}$ |
| HOMEPAP | $75.25_{\pm0.50}$ | $76.47_{\pm0.63}$ | $\mathbf{76.83_{\pm0.61}}$ | $76.13_{\pm0.93}$ |
| MASS | $70.00_{\pm1.91}$ | $71.52_{\pm1.13}$ | $\mathbf{72.77_{\pm1.09}}$ | $69.11_{\pm1.51}$ |
| MROS | $80.37_{\pm0.20}$ | $80.28_{\pm0.21}$ | $80.26_{\pm0.11}$ | $\mathbf{80.50_{\pm0.75}}$ |
| PhysioNet | $\mathbf{75.81_{\pm0.13}}$ | $74.68_{\pm0.55}$ | $74.82_{\pm2.11}$ | $74.58_{\pm1.57}$ |
| SHHS | $76.44_{\pm0.92}$ | $78.68_{\pm0.37}$ | $\mathbf{78.88_{\pm0.68}}$ | $78.77_{\pm0.67}$ |
| SOF | $81.08_{\pm1.14}$ | $80.68_{\pm1.38}$ | $79.49_{\pm0.41}$ | $80.10_{\pm0.62}$ |
| Mean | $77.55_{\pm0.34}$ | $77.78_{\pm0.46}$ | $\mathbf{78.34_{\pm0.42}}$ | $77.79_{\pm0.30}$ |

The main paper presents a critical difference diagram for the CNNTransformer evaluated on datasets balanced@400. It shows that PSDNorm with $F = 5$ is the best-performing normalization layer.

In Table 7, we report the results of different normalization layers used in the CNNTransformer architecture on datasets balanced@400.

Table 7: Different normalization layers used in the CNNTransformer architecture for datasets balanced@400.

| Dataset | BatchNorm | InstanceNorm | TMA | PSDNorm |
|---|---|---|---|---|
| ABC | $76.99_{\pm 0.53}$ | $75.40_{\pm 0.36}$ | $\mathbf{77.50_{\pm 0.54}}$ | $76.31_{\pm 0.46}$ |
| CCSHS | $86.75_{\pm 0.48}$ | $\mathbf{87.00_{\pm 0.34}}$ | $86.73_{\pm 0.25}$ | $86.92_{\pm 0.32}$ |
| CFS | $83.32_{\pm 0.35}$ | $\mathbf{83.77_{\pm 0.34}}$ | $83.16_{\pm 0.38}$ | $83.71_{\pm 0.29}$ |
| CHAT | $66.44_{\pm 0.49}$ | $66.40_{\pm 2.55}$ | $66.47_{\pm 1.37}$ | $\mathbf{70.04_{\pm 0.37}}$ |
| HOMEPAP | $74.81_{\pm 1.36}$ | $\mathbf{75.92_{\pm 0.44}}$ | $74.76_{\pm 0.83}$ | $75.26_{\pm 0.55}$ |
| MASS | $71.51_{\pm 0.47}$ | $71.70_{\pm 1.17}$ | $70.57_{\pm 0.80}$ | $\mathbf{72.55_{\pm 0.81}}$ |
| MROS | $79.77_{\pm 0.31}$ | $79.74_{\pm 0.55}$ | $\mathbf{79.85_{\pm 0.08}}$ | $79.77_{\pm 0.30}$ |
| PhysioNet | $72.54_{\pm 0.34}$ | $74.36_{\pm 0.84}$ | $71.39_{\pm 1.38}$ | $\mathbf{74.95_{\pm 0.41}}$ |
| SHHS | $75.34_{\pm 0.34}$ | $76.55_{\pm 0.92}$ | $75.15_{\pm 0.98}$ | $\mathbf{77.26_{\pm 0.57}}$ |
| SOF | $80.63_{\pm 0.60}$ | $80.78_{\pm 0.54}$ | $\mathbf{81.03_{\pm 0.48}}$ | $80.31_{\pm 0.90}$ |
| Mean | $76.38_{\pm 0.17}$ | $77.07_{\pm 0.28}$ | $76.30_{\pm 0.57}$ | $\mathbf{77.83_{\pm 0.36}}$ |

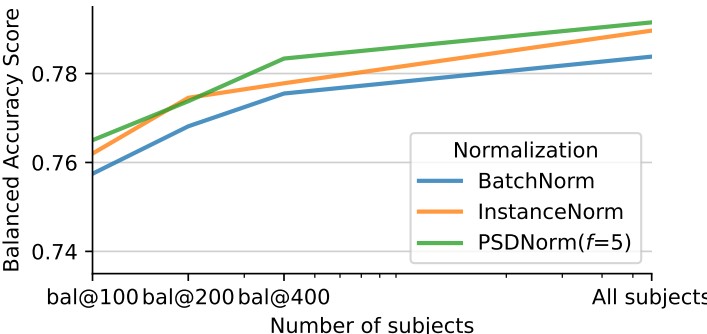

Figure 7: **Performance of PSDNorm and BatchNorm with varying training set sizes.** The BACC score is plotted against the number of training subjects used with U-Sleep.

First, we observe that CNNTransformer performs slightly below U-Sleep. Second, BatchNorm and InstanceNorm are the best performers on one and two datasets respectively, while PSDNorm achieves the best performance on 7 out of 10 datasets.

PSDNorm with $F = 5$ outperforms BatchNorm by a margin of 0.9 and InstanceNorm by 0.54 in average score.

These results highlight that PSDNorm is a plug-and-play normalization layer that can be seamlessly integrated into various architectures to reduce feature space variability.

## A.10 EVOLUTION OF PERFORMANCE WITH TRAINING SET SIZE

The choice of $f$ in PSDNorm controls the intensity of the normalization: larger $f$ provide stronger normalization, while smaller $f$ allow more flexibility in the model. In Figure 7, we evaluate its impact across different training set sizes and observe a clear trend: when trained on fewer subjects, larger filter sizes yield better performance (*i.e.,* $f = 17$), whereas smaller filter sizes are more effective with larger datasets (*i.e.,* $f = 5$). This suggests that with limited data, stronger normalization helps prevent overfitting, while with more data, a more flexible model is preferred. On average, PSDNorm with $f = 5$ offers a good compromise, achieving one of the best performances across all training set sizes.

## A.11 CRITICAL DIFFERENCE DIAGRAM FOR U-SLEEP ON ALL SUBJECTS

The main paper presents the critical difference diagram for U-Sleep on the dataset balanced@400. Figure 8 extends this analysis to all subjects across datasets. The conclusion remains consistent: PSDNorm with $F = 5$ is the best-performing normalization layer, while BatchNorm performs the

worst. Interestingly, PSDNorm with $F = 17$ ranks second to last, suggesting that overly strong adaptation can hurt performance when the dataset is large.

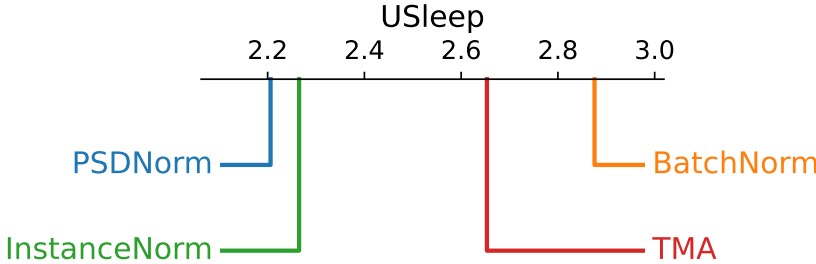

Figure 8: Critical difference diagram for U-Sleep on all subjects.

### A.12 COMPUTATIONAL TIME OF PSDNORM

Table 8: **Computational time of PSDNorm compared to BatchNorm and InstanceNorm for USleep and CNNTransformer.** The time is done for leave out the dataset CHAT and with the dataset balanced @400. The time is averaged over 3 runs and reported in seconds.

| Model | Normalization | Time per epoch (sec) | Time of inference (sec) |
|---|---|---|---|
| USleep | BatchNorm | $\mathbf{161.94 \pm 10.18}$ | $95.63 \pm 4.85$ |
| USleep | InstanceNorm | $258.71 \pm 2.15(*)$ | $\mathbf{93.40 \pm 8.18}$ |
| USleep | PSDNorm($f = 5$) | $172.85 \pm 4.05$ | $98.72 \pm 12.09$ |
| CNNTransformer | BatchNorm | $130.88 \pm 2.67$ | $93.03 \pm 5.25$ |
| CNNTransformer | InstanceNorm | $\mathbf{127.47 \pm 5.14}$ | $92.70 \pm 4.11$ |
| CNNTransformer | PSDNorm($f = 5$) | $152.83 \pm 2.29$ | $\mathbf{92.57 \pm 2.64}$ |

One important aspect of normalization layers is their computational cost, which can impact training and inference times. Table 8 compares the computational time of PSDNorm with BatchNorm and InstanceNorm in both U-Sleep and CNNTransformer architectures. In U-Sleep, PSDNorm takes 172.85 seconds per epoch, which is slightly higher than BatchNorm (161.94 seconds) but significantly lower than InstanceNorm (258.71 seconds). The high cost of InstanceNorm is due to the fact that the torch.compile was not working for Usleep and InstanceNorm. For inference, PSDNorm takes 98.72 seconds, which is comparable to BatchNorm (95.63 seconds) but slightly higher than InstanceNorm (93.40 seconds).

In CNNTransformer, PSDNorm takes 152.83 seconds per epoch, which is higher than BatchNorm (130.88 seconds) and InstanceNorm (127.47 seconds). However, for inference, PSDNorm is equivalent to both BatchNorm and InstanceNorm. The modest computational overhead introduced by PSDNorm is a worthwhile trade-off for its superior performance. This efficiency is enabled by the highly optimized implementation of the Fast Fourier Transform (FFT) on GPUs.

### A.13 CLASS-WISE PERFORMANCE

The Tables show that the most complicated sleep stage to classify is N1, with F1 scores consistently lower than other stages across all normalization methods. This is likely due to the inherent difficulty of distinguishing N1 from other stages, as it shares characteristics with both wakefulness and deeper sleep stages. In contrast, stages like Wake and REM tend to have higher F1 scores, indicating that they are easier to classify accurately.

Figure 9 illustrates the class-wise F1 score differences between normalization layers score against BatchNorm score. For almost all the classes, other normalization increase the performance compared to BatchNorm except for N1 where InstanceNorm shows a decrease in performance. PSDNorm is consistently the best performing normalization across all classes, highlighting its effectiveness in

Table 9: Class-wise F1 scores for BatchNorm layer on datasets balanced @ 400.

| Dataset | Wake | N1 | N2 | N3 | REM | F1 |
|---|---|---|---|---|---|---|
| ABC | $86.07_{\pm0.68}$ | $53.97_{\pm0.36}$ | $80.05_{\pm0.52}$ | $71.35_{\pm1.45}$ | $88.43_{\pm0.13}$ | $79.80_{\pm0.34}$ |
| CCSHS | $95.22_{\pm0.46}$ | $48.08_{\pm2.87}$ | $84.88_{\pm0.81}$ | $86.13_{\pm1.01}$ | $88.54_{\pm0.93}$ | $88.32_{\pm0.49}$ |
| CFS | $94.64_{\pm0.10}$ | $42.69_{\pm0.39}$ | $82.27_{\pm0.75}$ | $77.20_{\pm0.30}$ | $86.94_{\pm0.34}$ | $87.01_{\pm0.18}$ |
| CHAT | $78.35_{\pm1.31}$ | $35.57_{\pm2.64}$ | $52.63_{\pm2.77}$ | $72.67_{\pm1.25}$ | $76.12_{\pm1.50}$ | $66.56_{\pm1.42}$ |
| HOMEPAP | $84.51_{\pm1.01}$ | $41.45_{\pm0.71}$ | $73.88_{\pm2.00}$ | $57.41_{\pm1.72}$ | $82.52_{\pm1.33}$ | $76.20_{\pm1.25}$ |
| MASS | $66.93_{\pm4.65}$ | $40.02_{\pm2.00}$ | $78.73_{\pm0.67}$ | $67.01_{\pm0.39}$ | $75.80_{\pm5.65}$ | $76.06_{\pm1.69}$ |
| MROS | $94.63_{\pm0.25}$ | $41.73_{\pm1.11}$ | $73.74_{\pm0.72}$ | $47.86_{\pm0.33}$ | $82.70_{\pm0.46}$ | $83.69_{\pm0.39}$ |
| PhysioNet | $89.22_{\pm0.49}$ | $46.01_{\pm0.90}$ | $73.93_{\pm3.04}$ | $55.04_{\pm1.56}$ | $77.30_{\pm0.61}$ | $76.26_{\pm1.27}$ |
| SHHS | $85.68_{\pm1.57}$ | $32.56_{\pm1.34}$ | $72.48_{\pm2.00}$ | $61.91_{\pm1.31}$ | $79.09_{\pm1.07}$ | $76.98_{\pm0.70}$ |
| SOF | $93.91_{\pm0.15}$ | $38.29_{\pm1.09}$ | $79.15_{\pm0.57}$ | $71.86_{\pm3.17}$ | $86.49_{\pm0.18}$ | $85.49_{\pm0.58}$ |
| Mean | $86.91_{\pm1.07}$ | $42.04_{\pm1.34}$ | $75.17_{\pm1.39}$ | $66.84_{\pm1.25}$ | $82.39_{\pm1.22}$ | $79.64_{\pm0.83}$ |

Table 10: Class-wise F1 scores for LayerNorm layer with $f = 5$ on datasets balanced @ 400.

| Dataset | Wake | N1 | N2 | N3 | REM | F1 |
|---|---|---|---|---|---|---|
| ABC | $83.29_{\pm2.94}$ | $52.93_{\pm0.96}$ | $78.07_{\pm1.04}$ | $68.91_{\pm2.93}$ | $85.05_{\pm0.81}$ | $77.86_{\pm0.80}$ |
| CCSHS | $93.66_{\pm0.40}$ | $40.68_{\pm1.11}$ | $84.80_{\pm1.28}$ | $86.46_{\pm0.82}$ | $85.39_{\pm0.62}$ | $87.22_{\pm0.51}$ |
| CFS | $93.78_{\pm0.50}$ | $38.56_{\pm2.07}$ | $81.15_{\pm0.11}$ | $75.39_{\pm1.77}$ | $83.68_{\pm0.90}$ | $85.61_{\pm0.16}$ |
| CHAT | $72.07_{\pm2.84}$ | $30.11_{\pm0.63}$ | $46.82_{\pm6.98}$ | $70.45_{\pm2.92}$ | $68.11_{\pm4.07}$ | $61.32_{\pm2.25}$ |
| HOMEPAP | $83.58_{\pm1.72}$ | $43.85_{\pm1.55}$ | $73.97_{\pm1.76}$ | $57.54_{\pm1.92}$ | $79.16_{\pm1.19}$ | $76.15_{\pm1.13}$ |
| MASS | $60.73_{\pm4.02}$ | $40.56_{\pm0.87}$ | $77.39_{\pm6.28}$ | $65.91_{\pm4.01}$ | $69.92_{\pm10.53}$ | $73.95_{\pm5.80}$ |
| MROS | $94.12_{\pm0.10}$ | $38.41_{\pm2.21}$ | $71.48_{\pm3.66}$ | $47.85_{\pm0.37}$ | $79.05_{\pm0.66}$ | $82.22_{\pm1.27}$ |
| PhysioNet | $88.04_{\pm0.37}$ | $44.72_{\pm2.03}$ | $64.53_{\pm1.93}$ | $48.49_{\pm0.54}$ | $68.17_{\pm7.15}$ | $70.40_{\pm0.14}$ |
| SHHS | $84.61_{\pm2.49}$ | $32.89_{\pm1.92}$ | $72.02_{\pm2.48}$ | $61.13_{\pm1.59}$ | $75.63_{\pm0.63}$ | $75.98_{\pm0.22}$ |
| SOF | $93.24_{\pm0.27}$ | $35.61_{\pm2.49}$ | $77.49_{\pm2.15}$ | $70.83_{\pm3.09}$ | $83.52_{\pm0.01}$ | $84.23_{\pm1.30}$ |
| Mean | $84.71_{\pm1.56}$ | $39.83_{\pm1.58}$ | $72.77_{\pm2.77}$ | $65.30_{\pm1.99}$ | $77.77_{\pm2.66}$ | $77.49_{\pm1.36}$ |

Table 11: Class-wise F1 scores for Instancenorm layer on datasets balanced @ 400.

| Dataset | Wake | N1 | N2 | N3 | REM | F1 |
|---|---|---|---|---|---|---|
| ABC | $86.46_{\pm1.75}$ | $54.45_{\pm0.78}$ | $75.68_{\pm2.16}$ | $70.75_{\pm0.51}$ | $88.60_{\pm0.75}$ | $78.36_{\pm1.20}$ |
| CCSHS | $95.69_{\pm0.14}$ | $49.23_{\pm2.62}$ | $85.29_{\pm0.89}$ | $85.98_{\pm0.68}$ | $89.23_{\pm0.91}$ | $88.73_{\pm0.52}$ |
| CFS | $94.95_{\pm0.19}$ | $44.97_{\pm1.34}$ | $83.30_{\pm0.60}$ | $77.17_{\pm0.39}$ | $87.45_{\pm0.38}$ | $87.62_{\pm0.27}$ |
| CHAT | $77.67_{\pm6.38}$ | $29.47_{\pm8.45}$ | $48.78_{\pm6.21}$ | $71.34_{\pm2.64}$ | $74.58_{\pm1.83}$ | $64.19_{\pm4.63}$ |
| HOMEPAP | $86.27_{\pm0.53}$ | $43.19_{\pm1.46}$ | $75.19_{\pm1.07}$ | $58.39_{\pm1.03}$ | $83.44_{\pm0.46}$ | $77.66_{\pm0.58}$ |
| MASS | $67.25_{\pm1.95}$ | $43.19_{\pm2.05}$ | $78.64_{\pm1.76}$ | $65.78_{\pm1.37}$ | $78.34_{\pm1.18}$ | $76.94_{\pm1.12}$ |
| MROS | $94.80_{\pm0.18}$ | $41.46_{\pm0.71}$ | $74.42_{\pm1.16}$ | $48.89_{\pm2.31}$ | $82.11_{\pm0.10}$ | $83.95_{\pm0.53}$ |
| PhysioNet | $89.43_{\pm0.41}$ | $44.35_{\pm0.62}$ | $68.91_{\pm2.82}$ | $51.05_{\pm1.32}$ | $77.35_{\pm0.95}$ | $73.84_{\pm0.93}$ |
| SHHS | $88.62_{\pm0.30}$ | $33.02_{\pm2.26}$ | $74.31_{\pm2.07}$ | $64.28_{\pm0.72}$ | $80.32_{\pm0.42}$ | $79.12_{\pm0.96}$ |
| SOF | $94.42_{\pm0.20}$ | $37.18_{\pm2.37}$ | $78.43_{\pm1.88}$ | $72.39_{\pm1.56}$ | $86.82_{\pm0.70}$ | $85.50_{\pm0.86}$ |
| Mean | $87.55_{\pm1.20}$ | $42.05_{\pm2.27}$ | $74.29_{\pm2.06}$ | $66.60_{\pm1.25}$ | $82.83_{\pm0.77}$ | $79.59_{\pm1.16}$ |

Table 12: Class-wise F1 scores for TMA preprocessing with $f = 5$ on datasets balanced @ 400.

| Dataset | Wake | N1 | N2 | N3 | REM | F1 |
|---|---|---|---|---|---|---|
| ABC | $85.50_{\pm0.93}$ | $54.76_{\pm0.88}$ | $78.91_{\pm1.27}$ | $71.40_{\pm1.23}$ | $88.43_{\pm0.47}$ | $79.49_{\pm0.68}$ |
| CCSHS | $95.51_{\pm0.31}$ | $48.72_{\pm2.07}$ | $85.02_{\pm1.09}$ | $85.37_{\pm1.33}$ | $89.33_{\pm0.29}$ | $88.47_{\pm0.62}$ |
| CFS | $94.75_{\pm0.26}$ | $43.28_{\pm2.27}$ | $83.06_{\pm0.79}$ | $77.38_{\pm0.28}$ | $87.17_{\pm0.36}$ | $87.37_{\pm0.44}$ |
| CHAT | $80.70_{\pm4.88}$ | $37.51_{\pm1.85}$ | $58.89_{\pm4.08}$ | $75.71_{\pm2.09}$ | $75.95_{\pm2.35}$ | $69.90_{\pm2.74}$ |
| HOMEPAP | $84.13_{\pm0.98}$ | $43.92_{\pm0.51}$ | $74.83_{\pm1.40}$ | $57.94_{\pm1.19}$ | $81.58_{\pm0.13}$ | $76.83_{\pm0.97}$ |
| MASS | $70.45_{\pm7.07}$ | $41.83_{\pm3.63}$ | $77.30_{\pm1.48}$ | $65.04_{\pm2.01}$ | $79.50_{\pm2.42}$ | $76.32_{\pm0.36}$ |
| MROS | $94.50_{\pm0.32}$ | $41.74_{\pm1.13}$ | $75.20_{\pm1.26}$ | $48.92_{\pm2.07}$ | $82.35_{\pm0.70}$ | $84.15_{\pm0.46}$ |
| PhysioNet | $89.40_{\pm0.50}$ | $44.65_{\pm2.19}$ | $71.13_{\pm5.36}$ | $51.61_{\pm3.94}$ | $80.06_{\pm1.02}$ | $75.24_{\pm2.72}$ |
| SHHS | $88.30_{\pm0.58}$ | $32.95_{\pm2.17}$ | $73.23_{\pm2.58}$ | $62.33_{\pm0.33}$ | $79.27_{\pm0.84}$ | $78.19_{\pm0.90}$ |
| SOF | $93.69_{\pm0.34}$ | $37.85_{\pm2.16}$ | $79.33_{\pm1.49}$ | $72.46_{\pm1.92}$ | $86.83_{\pm0.29}$ | $85.56_{\pm0.90}$ |
| Mean | $87.69_{\pm1.62}$ | $42.72_{\pm1.89}$ | $75.69_{\pm2.08}$ | $66.82_{\pm1.64}$ | $83.05_{\pm0.89}$ | $80.15_{\pm1.08}$ |

Table 13: Class-wise F1 scores for PSDNorm layer with $f = 5$ on datasets balanced @ 400.

| Dataset | Wake | N1 | N2 | N3 | REM | F1 |
|---|---|---|---|---|---|---|
| ABC | $84.57_{\pm1.39}$ | $54.46_{\pm0.59}$ | $75.94_{\pm1.14}$ | $70.73_{\pm1.00}$ | $88.19_{\pm0.23}$ | $78.08_{\pm0.78}$ |
| CCSHS | $95.76_{\pm0.21}$ | $47.97_{\pm1.71}$ | $85.34_{\pm1.76}$ | $86.17_{\pm2.24}$ | $89.71_{\pm0.33}$ | $88.79_{\pm0.99}$ |
| CFS | $95.01_{\pm0.17}$ | $42.92_{\pm1.06}$ | $82.08_{\pm1.88}$ | $76.98_{\pm0.96}$ | $87.31_{\pm0.13}$ | $87.06_{\pm0.77}$ |
| CHAT | $82.93_{\pm1.99}$ | $36.59_{\pm6.79}$ | $61.84_{\pm2.98}$ | $77.06_{\pm1.32}$ | $78.01_{\pm1.62}$ | $71.86_{\pm0.95}$ |
| HOMEPAP | $85.75_{\pm1.45}$ | $44.47_{\pm0.78}$ | $75.54_{\pm1.72}$ | $58.70_{\pm1.40}$ | $83.24_{\pm0.55}$ | $77.85_{\pm1.29}$ |
| MASS | $72.74_{\pm2.12}$ | $42.20_{\pm1.16}$ | $78.56_{\pm2.98}$ | $66.13_{\pm2.37}$ | $78.23_{\pm3.21}$ | $77.16_{\pm1.73}$ |
| MROS | $94.63_{\pm0.31}$ | $41.40_{\pm1.64}$ | $73.33_{\pm1.80}$ | $47.56_{\pm2.25}$ | $82.52_{\pm0.47}$ | $83.51_{\pm0.84}$ |
| PhysioNet | $89.48_{\pm0.44}$ | $44.33_{\pm1.43}$ | $67.67_{\pm6.16}$ | $49.37_{\pm4.91}$ | $79.23_{\pm1.21}$ | $73.51_{\pm3.05}$ |
| SHHS | $89.09_{\pm0.66}$ | $33.78_{\pm2.53}$ | $74.15_{\pm2.70}$ | $64.40_{\pm0.29}$ | $80.24_{\pm1.03}$ | $79.26_{\pm1.35}$ |
| SOF | $93.74_{\pm0.27}$ | $34.70_{\pm2.45}$ | $76.58_{\pm2.97}$ | $70.20_{\pm2.02}$ | $85.50_{\pm1.63}$ | $84.14_{\pm1.05}$ |
| Mean | $88.37_{\pm0.90}$ | $42.28_{\pm2.01}$ | $75.10_{\pm2.61}$ | $66.73_{\pm1.88}$ | $83.22_{\pm1.04}$ | $80.12_{\pm1.28}$ |

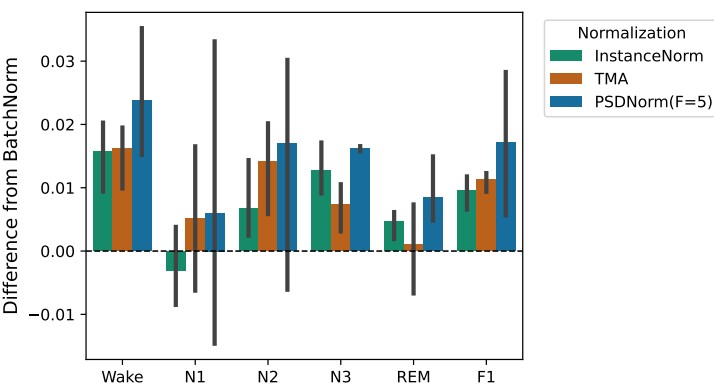

Figure 9: Class-wise F1 score differences between normalization layers. The variance is giving by the seeds.

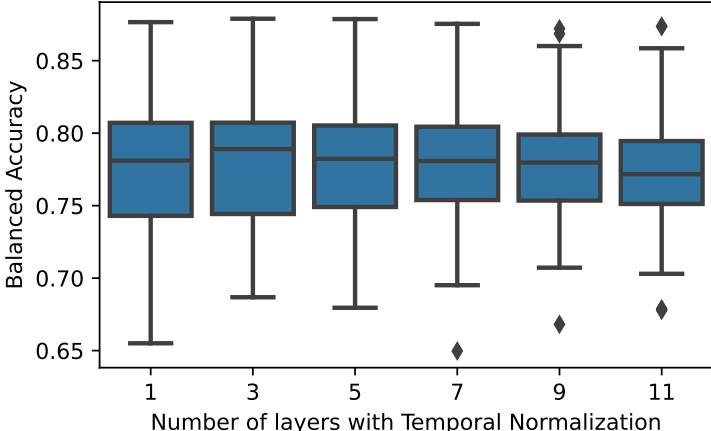

Figure 10: Impact of the number of layers in U-Sleep using PSDNorm with $f = 5$. The BACC score is plotted against the number of layers. The variance is one the Datasets.

Table 14: Performance of different normalization layers in U-Sleep in very low data regime on datasets balanced @ 40.

| Dataset | BatchNorm | LayerNorm | InstanceNorm | PSDNorm(F=5) | PSDNorm(F=15) |
|---|---|---|---|---|---|
| ABC | $\mathbf{73.83}_{\pm\mathbf{1.80}}$ | $64.54_{\pm3.46}$ | $72.29_{\pm1.50}$ | $71.35_{\pm1.15}$ | $72.61_{\pm1.84}$ |
| CCSHS | $83.58_{\pm0.45}$ | $77.91_{\pm1.59}$ | $\mathbf{85.33}_{\pm\mathbf{0.69}}$ | $85.10_{\pm0.17}$ | $85.00_{\pm0.25}$ |
| CFS | $81.13_{\pm0.85}$ | $76.57_{\pm1.89}$ | $81.59_{\pm0.47}$ | $\mathbf{81.79}_{\pm\mathbf{0.82}}$ | $80.93_{\pm0.13}$ |
| CHAT | $55.74_{\pm2.38}$ | $58.42_{\pm1.64}$ | $63.38_{\pm5.26}$ | $59.66_{\pm0.89}$ | $\mathbf{67.86}_{\pm\mathbf{3.59}}$ |
| HOMEPAP | $74.52_{\pm1.66}$ | $72.19_{\pm1.65}$ | $76.03_{\pm0.48}$ | $76.01_{\pm0.32}$ | $\mathbf{76.14}_{\pm\mathbf{1.63}}$ |
| MASS | $\mathbf{70.15}_{\pm\mathbf{3.09}}$ | $64.79_{\pm2.72}$ | $66.58_{\pm0.30}$ | $69.49_{\pm1.12}$ | $68.21_{\pm6.25}$ |
| MROS | $77.12_{\pm0.03}$ | $71.76_{\pm2.38}$ | $76.59_{\pm0.28}$ | $\mathbf{77.19}_{\pm\mathbf{0.38}}$ | $76.77_{\pm1.30}$ |
| PhysioNet | $71.68_{\pm1.61}$ | $69.59_{\pm1.46}$ | $72.68_{\pm3.09}$ | $\mathbf{73.67}_{\pm\mathbf{0.93}}$ | $72.08_{\pm3.65}$ |
| SHHS | $73.74_{\pm1.11}$ | $71.50_{\pm0.97}$ | $75.56_{\pm1.66}$ | $75.43_{\pm1.38}$ | $\mathbf{76.00}_{\pm\mathbf{0.63}}$ |
| SOF | $75.84_{\pm2.16}$ | $73.50_{\pm1.97}$ | $\mathbf{76.54}_{\pm\mathbf{1.59}}$ | $75.14_{\pm1.79}$ | $76.00_{\pm3.00}$ |
| Mean | $73.35_{\pm0.87}$ | $70.72_{\pm1.22}$ | $75.19_{\pm1.03}$ | $74.79_{\pm0.75}$ | $\mathbf{75.88}_{\pm\mathbf{0.93}}$ |

improving sleep stage classification. But we have to note that for N1 and N2 the variance over the seed is big showing the instability of the training for these classes.

## A.14 Study of impact of number of layer in U-Sleep using PSDNorm

In the main paper, we apply PSDNorm in 3 layers of U-Sleep. Here, we investigate the impact of varying the number of layers that utilize PSDNorm. Figure 10 shows the BACC score as a function of the number of layers with PSDNorm. The results indicate that increasing the number of layers with PSDNorm reach a plateau after 3 layers, but does reduce the variance across datasets. This suggests that while adding more layers with PSDNorm can enhance performance, there are diminishing returns beyond a certain point. Thus, using PSDNorm in 3 layers strikes a good balance between performance, good variance, and computational efficiency.

## A.15 Study in very low data regime

In this section, we explore the performance of different normalization layers in U-Sleep when trained on a very limited dataset, specifically balanced @ 40 subjects. The results indicate that in this low data regime, PSDNorm with a small filter size ($F = 5$) struggle to outperform InstanceNorm while still outperforming BatchNorm and LayerNorm. However, PSDNorm with a larger filter size ($F = 15$) gives the best average performance across datasets with an increase of more than 10% in BACC

Table 15: **Comparison of PSDNorm with AdaBN on datasets balanced @400 using U-Sleep.**

| Dataset | BatchNorm | LayerNorm | InstanceNorm | AdaBN(3) | AdaBN(12) | AdaBN(full) | TMA | PSDNorm |
|---------|-----------|-----------|--------------|----------|-----------|-------------|-----|---------|
| ABC | $78.26_{\pm 1.33}$ | $75.29_{\pm 0.81}$ | $\mathbf{78.73_{\pm 0.42}}$ | $78.25_{\pm 1.30}$ | $77.21_{\pm 1.56}$ | $76.89_{\pm 1.30}$ | $78.04_{\pm 0.51}$ | $78.18_{\pm 0.68}$ |
| CCSHS | $87.42_{\pm 0.16}$ | $85.20_{\pm 0.48}$ | $\mathbf{87.62_{\pm 0.42}}$ | $87.38_{\pm 0.17}$ | NaN | $86.99_{\pm nan}$ | $87.57_{\pm 0.20}$ | $87.58_{\pm 0.30}$ |
| CFS | $84.32_{\pm 0.57}$ | $81.66_{\pm 1.36}$ | $\mathbf{84.72_{\pm 0.33}}$ | $84.21_{\pm 0.60}$ | NaN | $83.61_{\pm nan}$ | $84.58_{\pm 0.20}$ | $84.29_{\pm 0.36}$ |
| CHAT | $66.55_{\pm 0.88}$ | $61.19_{\pm 1.16}$ | $64.43_{\pm 4.41}$ | $66.49_{\pm 0.89}$ | NaN | NaN | $68.73_{\pm 2.48}$ | $\mathbf{70.28_{\pm 1.70}}$ |
| HOMEPAP | $75.25_{\pm 0.50}$ | $74.86_{\pm 0.25}$ | $76.47_{\pm 0.63}$ | $75.15_{\pm 0.46}$ | $74.39_{\pm 0.56}$ | $74.46_{\pm 0.53}$ | $76.10_{\pm 0.32}$ | $\mathbf{76.83_{\pm 0.61}}$ |
| MASS | $70.00_{\pm 1.91}$ | $68.56_{\pm 3.33}$ | $71.52_{\pm 1.13}$ | $69.68_{\pm 1.66}$ | $68.46_{\pm 2.58}$ | $68.31_{\pm 1.86}$ | $71.63_{\pm 1.92}$ | $\mathbf{72.77_{\pm 1.09}}$ |
| MROS | $\mathbf{80.37_{\pm 0.20}}$ | $78.05_{\pm 0.22}$ | $80.28_{\pm 0.21}$ | $80.34_{\pm 0.20}$ | NaN | NaN | $80.09_{\pm 0.40}$ | $80.26_{\pm 0.11}$ |
| PhysioNet | $\mathbf{75.81_{\pm 0.13}}$ | $71.82_{\pm 2.12}$ | $74.68_{\pm 0.55}$ | $75.27_{\pm 0.14}$ | $74.01_{\pm 0.13}$ | $74.01_{\pm 0.14}$ | $75.31_{\pm 1.54}$ | $74.82_{\pm 2.11}$ |
| SHHS | $76.44_{\pm 0.92}$ | $75.12_{\pm 0.39}$ | $78.68_{\pm 0.37}$ | $76.43_{\pm 0.92}$ | NaN | NaN | $77.00_{\pm 0.39}$ | $\mathbf{78.88_{\pm 0.68}}$ |
| SOF | $81.08_{\pm 1.14}$ | $78.70_{\pm 0.50}$ | $80.68_{\pm 1.38}$ | $81.05_{\pm 1.13}$ | NaN | $81.17_{\pm nan}$ | $\mathbf{81.25_{\pm 0.71}}$ | $79.49_{\pm 0.41}$ |
| Mean | $77.22_{\pm 0.34}$ | $75.04_{\pm 0.42}$ | $78.17_{\pm 0.28}$ | $77.18_{\pm 0.34}$ | $74.29_{\pm 1.08}$ | $76.59_{\pm 4.82}$ | $77.74_{\pm 0.36}$ | $\mathbf{78.85_{\pm 0.59}}$ |

compared to other normalization layers for CHAT dataset. This suggests that in scenarios with very limited data, stronger normalization (larger filter size) is beneficial to prevent overfitting and enhance generalization.

### A.16 COMPARISON WITH ADABN

Table 15 presents a comparison between PSDNorm and AdaBN using the U-Sleep architecture on datasets balanced @400. AdaBN adapts the BatchNorm statistics separately for each subject. In the original paper, all BN layers are replaced (AdaBN(full)), but for a fair comparison we also evaluate two additional settings: AdaBN(3), which adapts only the first three BN layers, and AdaBN(12), which adapts only the first BN layers of the encoders.

As expected, AdaBN struggles to achieve strong performance on sleep staging. It consistently underperforms compared to TMA and, in some cases, even performs worse than standard BatchNorm. Notably, increasing the number of adapted BN layers further degrades performance, highlighting the importance of not adapting too many layers within the model. In contrast, PSDNorm consistently outperforms AdaBN across all datasets, demonstrating its effectiveness in normalizing features for sleep staging tasks.

