# OpenReview forum: "PSDNorm: Temporal Normalization for Deep Learning in Sleep Staging"
_ICLR.cc/2026/Conference — ICLR 2026 Poster_

### Official Review · Reviewer_6ujs · 2025-10-24

**Soundness:** 3
**Presentation:** 3
**Contribution:** 2
**Rating:** 6
**Confidence:** 4

**Summary:**

The paper proposes PSDNorm (Power Spectral Density Normalization), a normalization layer that can be directly inserted into a network for time-series learning. Its core idea is to perform "power spectrum alignment" on feature maps in the time-frequency domain, allowing the model to learn domain-invariant temporal statistical structures. Experiments on a large-scale sleep dataset demonstrate the effectiveness of the framework.

**Strengths:**

1. The structure of this paper is clear and reasonable.

2. This paper is easy to follow and generally well presented.

3. Extensive experiments have been conducted to evaluate the performance of the proposed framework.

**Weaknesses:**

1. The current related work section is more like a preliminary. There should be several subsections introducing works with similar topics, such as methods for training sleep data and normalization in time series.

2. The feature map of the middle layer of the deep network may not meet the prior conditions in the article after nonlinearity and convolution, a Gaussian periodic signal, a covariance block diagonal, and diagonalized according to the Fourier basis. Should we consider the failure mode?

3. The comparison is mainly limited to normalization layers, but the core idea of PSDNorm is based on Monge/OT/PSD alignment. It could be better to compare PSDNorm with existing TMA/STMA or general test-time adaptation methods. It could also compare PSDNorm with TMA, which uses PSD as preprocessing (or TMA embedded in the network).

4. It could be better to consider the parameter sensitivity of the Welch estimator.

5. It should show more results on different testing windows.

6. What about the computational cost of this normalization?

**Questions:**

See in weakness.

---

> ### Author Response · Authors · 2025-11-22
>
> >Weaknesses:
> The current related work section is more like a preliminary. There should be several subsections introducing works with similar topics, such as methods for training sleep data and normalization in time series.
>
> Thanks for the suggestion, we added in the beginning of the related work a paragraph that introduces deep learning for sleep staging with different possible architecture and a description of U-Time architecture that we use in the paper.
>
> > The feature map of the middle layer of the deep network may not meet the prior conditions in the article after nonlinearity and convolution, a Gaussian periodic signal, a covariance block diagonal, and diagonalized according to the Fourier basis.
> > Should we consider the failure mode?
>
>  As discussed in Section 3.3, our core assumption is that the distribution shift is primarily captured by the first- and second-order statistics, and we therefore use a Gaussian assumption to estimate this shift. We do not assume that the data itself is Gaussian, just as BatchNorm does not assume Gaussianity, yet estimates the mean and (diagonal) covariance from the data to correct for distributional shifts.
>
> As discussed above if the data contain discriminative information specifically encoded in the first- or second-order moments, PSDNorm may cancel this information and become less advantageous than alternative approaches. This consideration also motivates our choice to limit the number of PSDNorm layers in the network: beyond a certain depth, second-order statistics can carry crucial information for classification, and over-normalizing them could harm performance. We will amend the paper to further discuss this.
>
>
> >The comparison is mainly limited to normalization layers, but the core idea of PSDNorm is based on Monge/OT/PSD alignment. It could be better to compare PSDNorm with existing TMA/STMA or general test-time adaptation methods. It could also compare PSDNorm with TMA, which uses PSD as preprocessing (or TMA embedded in the network).
>
> We fully agree that comparing to TMA is important for a fair evaluation, which is why PSDNorm is already compared to TMA in Table 2. We also agree that comparisons to test-time adaptation methods could be interesting. It is important to note, however, that our method does not require any additional training at inference time. This limits the area of comparison because we can apply our model online on new data without requiring fine tuning on new data.
>
> As suggested by Reviewer 1, we compared PSDNorm to AdaBN and provide the results in Table 15 (and in the next comment). We include a comparison with AdaBN using the authors’ official implementation. We adapted the BatchNorm statistics separately for each subject. In the original paper, all BN layers are replaced (AdaBN(full)), but for a fair comparison we also evaluate one additional settings: AdaBN(3), which adapts only the first three BN layers.
>
> As expected, AdaBN struggles to achieve strong performance on sleep staging. It consistently underperforms compared to TMA and, in some cases, even performs worse than standard BatchNorm. Notably, increasing the number of adapted BN layers further degrades performance, highlighting the importance of not adapting too many layers within the model.
>
>
> >It could be better to consider the parameter sensitivity of the Welch estimator.
>
> The Welch method has few parameters. In our experiments, we used the default settings except for the FFT length which is set at the filter size f. A sensitivity analysis of this parameter is provided in the appendix.
>
> >It should show more results on different testing windows.
>
> Could you please clarify what you mean by testing windows? If you mean filter size, we already did a sensitivity analysis in Appendix, Figure 6.
>
> >What about the computational cost of this normalization?
>
> We agree that computational cost is an important consideration for a new layer. In Table 8 of the Appendix, we provide a comparison of computation times for different layers. PSDNorm increases training time by approximately 10%, which is negligible, especially considering that inference time remains unchanged.

---

> > ### Author Response · Authors · 2025-11-24
> >
> > | Dataset   | BatchNorm           | LayerNorm         | InstanceNorm                | AdaBN(3)           | AdaBN(full)        | TMA                  | PSDNorm              |
> > |-----------|----------------------|-------------------|-----------------------------|--------------------|--------------------|-----------------------|-----------------------------|
> > | ABC       | 78.26±1.33           | 75.29±0.81        | **78.73±0.42**              | 78.25±1.30         | 76.89±1.30         | 78.04±0.51            | 78.18±0.68                  |
> > | CCSHS     | 87.42±0.16           | 85.20±0.48        | **87.62±0.42**              | 87.38±0.17         | 87.02±0.16         | 87.57±0.20            | 87.58±0.30                  |
> > | CFS       | 84.32±0.57           | 81.66±1.36        | **84.72±0.33**              | 84.21±0.60         | 84.05±0.56         | 84.58±0.20            | 84.29±0.36                  |
> > | CHAT      | 66.55±0.88           | 61.19±1.16        | 64.43±4.41                  | 66.49±0.89         | 66.44±0.88         | 68.73±2.48            | **70.28±1.70**              |
> > | HOMEPAP   | 75.25±0.50           | 74.86±0.25        | 76.47±0.63                  | 75.15±0.46         | 74.46±0.53         | 76.10±0.32            | **76.83±0.61**              |
> > | MASS      | 70.00±1.91           | 68.56±3.33        | 71.52±1.13                  | 69.68±1.66         | 68.31±1.86         | 71.63±1.92            | **72.77±1.09**              |
> > | MROS      | **80.37±0.20**       | 78.05±0.22        | 80.28±0.21                  | 80.34±0.20         | 80.27±0.20         | 80.09±0.40            | 80.26±0.11                  |
> > | PhysioNet | **75.81±0.13**       | 71.82±2.12        | 74.68±0.55                  | 75.27±0.14         | 74.01±0.14         | 75.31±1.54            | 74.82±2.11                  |
> > | SHHS      | 76.44±0.92           | 75.12±0.39        | 78.68±0.37                  | 76.43±0.92         | 76.41±0.92         | 77.00±0.39            | **78.88±0.68**              |
> > | SOF       | 81.08±1.14           | 78.70±0.50        | 80.68±1.38                  | 81.05±1.13         | 80.67±1.13         | **81.25±0.71**         | 79.49±0.41                  |
> > | Mean      | 77.22±0.34           | 75.04±0.42        | 78.17±0.28                  | 77.18±0.34         | 77.08±0.34         | 77.74±0.36            | **78.85±0.59**              |

---

> > > ### Comment · Reviewer_6ujs · 2025-11-28
> > >
> > > Thank you for your detailed responses. Most of my concerns have been resolved. The W5's "testing windows" refers to the length of the time series slices in the test set. This is a minor point, where providing analysis would be helpful, but it's not mandatory.

---

> > > > ### Author Response · Authors · 2025-11-28
> > > >
> > > > Thank you for your positive follow-up. We appreciate your careful reading of our work and your constructive comments.
> > > >
> > > > In our experiments we used the size of test window of Usleep and we take the same size for training and testing to avoid an additional shift but we agree that this is an interesting question. We will not have time to perform experiments until the end of discussion but will work on it for camera ready.
> > > >
> > > > If you have any further concerns or suggestions, please feel free to let us know. We would be happy to clarify anything.

---

### Official Review · Reviewer_ovsu · 2025-10-30

**Soundness:** 3
**Presentation:** 3
**Contribution:** 3
**Rating:** 4
**Confidence:** 4

**Summary:**

This paper introduces PSDNorm, a novel normalization layer designed for deep learning on sequential signals. Unlike standard normalization layers (BatchNorm, LayerNorm, InstanceNorm) that treat time steps independently, PSDNorm leverages the temporal auto-correlation structure of signals by aligning their Power Spectral Density (PSD) to a running Riemannian barycenter via Monge mapping. The method is evaluated extensively on sleep stage classification using 10 datasets comprising ~10K subjects. Results demonstrate consistent improvements over baseline normalization techniques, with particular benefits in data-limited regimes and under domain shift.

**Strengths:**

1. LODO setup with 10 datasets, proper statistical testing, multiple seeds, and comprehensive ablations set a high standard. The transparency in reporting (including failed torch.compile) is commendable.

2. The use of optimal transport, Wasserstein barycenters, and Riemannian geometry provides solid theoretical foundation, even if the novelty is limited.

3. Data efficiency results (Figure 6, Table 2 bottom) demonstrate real value for resource-constrained scenarios common in medical applications.

4. Algorithm 1, architecture details (Table 4), implementation notes, and comprehensive appendix facilitate reproducibility.

**Weaknesses:**

1. Limited theoretical justification:
- When do we expect PSDNorm to outperform alternatives? No formal analysis provided.
- Why is the geodesic update (Eq. 6) optimal? Other update schemes not explored.
- Gaussian assumption (Eq. 2) likely violated for deep network features - no empirical validation.
- No analysis of what happens when periodicity assumption fails.

2. Narrow evaluation scope:
- Only sleep staging evaluated - no other time series tasks
- Both architectures are CNN-based - what about pure transformers or RNNs?
- No evaluation on non-biomedical sequential data
- Limits ability to assess general applicability


3. This paper might overclaimed contributions, "4× more data-efficient" is misleading, should say "more robust to data scarcity"; "State-of-the-art" improvements are marginal (often <1%); Title suggests "temporal normalization" but method is specific to signals with spectral content

4. Something missed
- No visualization of learned barycenter PSDs - are they interpretable?
- No analysis of what spectral patterns PSDNorm captures
- Missing failure case analysis
- No discussion of when PSDNorm might be worse than alternatives


5. Computational:
- Training time overhead 7-17% (Table 8) is non-negligible
- Requires FFT operations which may not be hardware-optimized on all platforms
- No analysis of memory overhead
- The InstanceNorm timing anomaly (torch.compile issue) undermines the comparison

**Questions:**

1. Can you empirically verify that intermediate feature maps exhibit the assumed covariance structure (Eq. 2)? Visualizations of PSD estimates from actual feature maps would strengthen the claims.
2. Under what conditions (signal properties, network architecture, data distribution) should we expect PSDNorm to outperform alternatives? Can you provide theoretical guidance?
3. What do the learned running barycenter PSDs look like? Are they interpretable? Do they differ meaningfully across layers or datasets?
4. Have you evaluated PSDNorm on other time series tasks (e.g., ECG, audio, speech)? Results on even 1-2 additional domains would strengthen generality claims.
5. Have you tried alternatives to the geodesic update (Eq. 6)? What about learnable momentum α or adaptive filter sizes f?

---

> ### Author Response · Authors · 2025-11-22
>
> >Weaknesses:
> Limited theoretical justification:
>
> We acknowledge that our work does not include a formal theoretical justification, although it is grounded in non-trivial optimal transport theory and subtle signal processing estimation concepts. Nonetheless, we note that the introduction of new neural network layers has historically emphasized empirical validation rather than theoretical analysis, and we believe it would be unreasonable to hold our contribution to a different standard.
>
>
> > Geodesic update:
> >- Have you tried alternatives to the geodesic update (Eq. 6)? What about learnable momentum α or adaptive filter sizes f?
> >- Why is the geodesic update (Eq. 6) optimal? Other update schemes not explored.
>
> We rely on the natural geometry induced by OT mappings and barycenters. Using this geodesic update is therefore a natural choice, leading to fast computation and strong performance. We did not explore other update scheme. While it would be interesting, we think it is beyond the scope of the current paper.
>
>
> >- Gaussian assumption (Eq. 2) likely violated for deep network features - no empirical validation.
> >- No analysis of what happens when periodicity assumption fails.
>
> As discussed in Section 3.3, our core assumption is that the distribution shift is primarily captured by the first- and second-order statistics, and we therefore use a Gaussian assumption to estimate this shift. We do not assume that the data itself is Gaussian—just as BatchNorm does not assume Gaussianity, yet estimates the mean and (diagonal) covariance from the data to correct for distributional shifts.
>
> > Failure mode:
> >- When do we expect PSDNorm to outperform alternatives? No formal analysis provided.
> >- Missing failure case analysis
> >- No discussion of when PSDNorm might be worse than alternatives
> >- Under what conditions (signal properties, network architecture, data distribution) should we expect PSDNorm to outperform alternatives? Can you provide theoretical guidance?
>
> As discussed above if the data contain discriminative information specifically encoded in the first- or second-order moments, PSDNorm may cancel this information and become less advantageous than alternative approaches. This consideration also motivates our choice to limit the number of PSDNorm layers in the network: beyond a certain depth, second-order statistics can carry crucial information for classification, and over-normalizing them could harm performance. We will amend the paper to further discuss this.
>
> >Narrow evaluation scope:
> >- Only sleep staging evaluated - no other time series tasks
> >- No evaluation on non-biomedical sequential data
> >- Limits ability to assess general applicability
> >- Have you evaluated PSDNorm on other time series tasks (e.g., ECG, audio, speech)? Results on even 1-2 additional domains would strengthen generality claims.
>
> We fully agree that incorporating additional modalities would be an interesting research direction. However, we have designed what we think are very advanced sleep staging experiments on SOTA models and with robust performance evaluation (Leave-One-Domain-Out) which correspond to substantial computational cost and require space to be presented properly. Adding an experiment on a new modality (ECG or MRI), would require similar work including state-of-the-art baselines that would require substantial work and ressources that is not available to us.
>
> >- Both architectures are CNN-based - what about pure transformers or RNNs?
>
> Transformers applied to signals or images typically begin with convolutional layers, often including a “patchification” step. Our CNN-Transformer does not perform patchification, thereby avoiding the need to select a patch size and instead relying only on stride and kernel width.
> Regarding RNNs, we acknowledge that bidirectional RNNs have been proposed for the sleep staging task; however, they are generally not found to outperform Transformers, which also effectively leverage temporal context. We will add a note in the paper discussing RNNs. We also agree that integrating PSDNorm into an RNN architecture would be less straightforward.

---

> ### Author Response · Authors · 2025-11-22
>
> >- This paper might overclaimed contributions, "4× more data-efficient" is misleading, should say "more robust to data scarcity";
>
> Thanks for suggesting this, we will change this claim in the introduction.
>
> >- "State-of-the-art" improvements are marginal (often <1%);
>
> In sleep staging, even expert annotators exhibit error rates of around 20% (see Table 3 in the USleep paper). As a result, achieving improvements of several percentage points is inherently difficult, since much of the variability arises from differences in interpretation among specialists.
>
> >- Title suggests "temporal normalization" but method is specific to signals with spectral content
>
> By "temporal normalization," we mean that the normalization is applied to the temporal content underlying the spectral information, as opposed to spatial normalization.
>
> > PSD visualization:
> >- No visualization of learned barycenter PSDs - are they interpretable? No analysis of what spectral patterns PSDNorm captures
> >- What do the learned running barycenter PSDs look like? Are they interpretable? Do they differ meaningfully across layers or datasets?
> >- Can you empirically verify that intermediate feature maps exhibit the assumed covariance structure (Eq. 2)? Visualizations of PSD estimates from actual feature maps would strengthen the claims.
>
> Thank you for the suggestion. We added a visualization of the PSD at different stages of the network. The results show that using BatchNorm or TMA is insufficient to address the shift that emerges inside the model. In contrast, normalization layers such as InstanceNorm or PSDNorm help realign the PSD throughout the network, highlighting the importance of normalizing beyond the input space.
>
> >Computational:
> >- Training time overhead 7-17% (Table 8) is non-negligible
>
> We agree that a 7% time overhead can be non-negligible; however, in this case it only affects training. In the current era of deep learning, a modest increase in training time is generally acceptable, especially since inference time, which is more critical for end users, is unaffected. Once the model is trained, no additional overhead is incurred during inference.
>
> > Requires FFT operations which may not be hardware-optimized on all platforms
>
> FFT is one of the most widely used algorithms, and many optimized CuFFT implementations already exist across hardware architectures.
>
> >- No analysis of memory overhead
>
> We did not include a memory-overhead analysis, memory usage is not a limiting factor for our method since statistics computed are very small.
>
> >- The InstanceNorm timing anomaly (torch.compile issue) undermines the comparison
>
> We agree that this issue complicates the comparison. However, PyTorch is known to be non-deterministic in several operations and in particular compiling, and this is not something we can fully control.

---

### Official Review · Reviewer_vTBu · 2025-10-31

**Soundness:** 3
**Presentation:** 3
**Contribution:** 3
**Rating:** 8
**Confidence:** 3

**Summary:**

This paper proposes a novel deep learning normalization method—PSDNorm (Temporal Normalization Layer)—aimed at addressing the data distribution bias problem in biomedical data (such as sleep staging). Traditional normalization methods (such as BatchNorm, LayerNorm, and InstanceNorm) typically ignore temporal correlations, while PSDNorm, by employing Monogram mapping and temporal context, can effectively normalize feature maps in deep learning models, thereby improving performance on unseen data.

**Strengths:**

1) The proposed PSDNorm layer effectively addresses the data distribution shift problem in biomedical signals within a deep learning framework, providing a novel perspective and approach for processing complex time-series signals.

2) Experimental results show that PSDNorm exhibits superior performance on multiple large-scale sleep datasets, particularly in handling data distribution shifts, outperforming existing normalization methods (such as BatchNorm and LayerNorm).

3) PSDNorm is a "plug-and-play" normalization layer that can be seamlessly integrated into existing neural network architectures with relatively low computational cost.

**Weaknesses:**

1) The proposed PSDNorm introduces a filter size hyperparameter (f), which controls the normalization strength. The choice of f can affect model performance across different datasets and tasks. Although the authors provide some default values, automatically selecting this parameter in adaptive settings may still be a challenge.

2) While the experiments cover multiple sleep datasets, the diversity of these datasets remains limited and may not fully represent all challenges in biomedical signals. Future research could consider extending this method to other domains, such as audio processing and other biomedical applications, to further validate its generalization capabilities.

**Questions:**

See the section on weaknesses for details.

---

> ### Author Response · Authors · 2025-11-22
>
> >The proposed PSDNorm introduces a filter size hyperparameter (f), which controls the normalization strength. The choice of f can affect model performance across different datasets and tasks. Although the authors provide some default values, automatically selecting this parameter in adaptive settings may still be a challenge.
>
> This is a fundamental question when dealing with distribution shifts, but also a complex one. Although domain-adaptation scoring methods exist, our setting offers a different opportunity: because we have many datasets available, we can in principle validate our hyperparameters by holding out one or two datasets as validation sets. We acknowledge that this analysis could have been considered in our paper.
>
> However, here we identified a parameter value that performs well across all datasets. Selecting the optimal parameter for each dataset and for each training-set size would likely yield even better results. This would nevertheless come at a significant computational cost which does not seem justified given the already available positive results.
>
> >While the experiments cover multiple sleep datasets, the diversity of these datasets remains limited and may not fully represent all challenges in biomedical signals. Future research could consider extending this method to other domains, such as audio processing and other biomedical applications, to further validate its generalization capabilities.
>
> We fully agree that incorporating additional modalities would be an interesting research direction. However, we have designed what we think are very advanced sleep staging experiments on SOTA models and with robust performance evaluation (Leave-One-Domain-Out) which correspond to substantial computational cost and require space to be presented properly. Adding an experiment on a new modality (ECG or MRI), would require similar work including state-of-the-art baselines that would require substantial work and resources that is not available to us.

---

### Official Review · Reviewer_yQGx · 2025-11-01

**Soundness:** 2
**Presentation:** 2
**Contribution:** 2
**Rating:** 4
**Confidence:** 3

**Summary:**

The paper introduces a new normalization method called PSDNorm to address the challenges that models applied to sleep staging face. The authors point out challenges in building neural networks for sleep staging, like distribution shift across subjects, devices, and datasets, and they point out limitations in existing normalization layers like BatchNorm/LayerNorm, which ignore temporal autocorrelation in signals. PSDNorm aligns the power spectral density of intermediate representations to a barycenter using an f-Monge mapping. The authors claim that they achieve SOTA generalization under a leave-one-dataset evaluation across 10 datasets, and they find that they are 4x more data efficient.

**Strengths:**

•	Principled Formulation: The optimal transport on Gaussian models is clean. I can understand why this normalization would be beneficial for OOD generalization.
•	Domain shift: Authors show that PSD has a strong gain, notably with MASS and CHAT, and on the hardest subjects, as shown in Figure 4.
•	Data Efficiency: If the hyperparameter is well-tuned, it seems that PSDNorm requires 4x fewer labeled subjects than BatchNorm.

**Weaknesses:**

I think this paper needs stronger comparisons in order to better characterize whether PSDNorm is making a real difference. I also think better scoping the claims will help significantly.

•	Normalization replacement: In my opinion, the paper makes a very odd choice in the experiments, as stated in line 386, with mixing normalization strategies. Instead of replacing the entire network, the paper takes a CNN Transformer and replaces the first three BatchNorm layers, keeping all other BatchNorm layers. For TMA, no layer is replaced, and this is treated as a preprocessing step, but TMA uses BatchNorm as well. This design makes comparisons confounded by remaining BatchNorm layers and by different amounts of architectural change across baselines. I think a full replacement is required here for clean comparison.
•	Channel independence: The method uses per-channel PSDs, from my understanding, as shown in Eq. 2. I think this assumes uncorrelated sensors. For learned feature maps, channel independence is unlikely to hold, so cross-channel correlations are ignored. I think this caps achievable alignment across channels.
•	Test-time adaptation: This point is more minor. The paper discusses TTA in Section 3.3. However, I think the comparison overlooks key issues with PSDNorm for TTA that make me uncertain about the claims in line 274. PSDNorm does not update its barycenter at inference (Algorithm 1), so it does not adapt to a novel target domain at test time.

**Questions:**

•	Hyperparameters: The paper fixes \alpha and f, with sensitivity analysis mainly on f. How important is \alpha? Other aspects like Welch window overlap or FFT length could be useful to discuss as well to get a sense of stability and cost.
•	EEG channels: Will PSDNorm actually scale to other channels or modalities?

---

> ### Author Response · Authors · 2025-11-22
>
> >• Normalization replacement: In my opinion, the paper makes a very odd choice in the experiments, as stated in line 386, with mixing normalization strategies. Instead of replacing the entire network, the paper takes a CNN Transformer and replaces the first three BatchNorm layers, keeping all other BatchNorm layers. For TMA, no layer is replaced, and this is treated as a preprocessing step, but TMA uses BatchNorm as well. This design makes comparisons confounded by remaining BatchNorm layers and by different amounts of architectural change across baselines. I think a full replacement is required here for clean comparison.
>
> PSDNorm reduces variability in the PSD space. Our assumption is that the distribution shift is largely captured by the lower-order moments. However, deeper in the architecture, these first-order statistics can carry important information for classification.
>
> We do not apply PSDNorm to the whole network because of the signal hierarchy. Early layers process non-stationary raw signals (requiring covariance alignment), while deep layers process abstract semantic features (where BatchNorm is sufficient).
>
> To investigate this trade-off, we include a sensitivity analysis on the number of layers using PSDNorm. The results show that applying PSDNorm to more than three layers does not improve accuracy, even though variance continues to decrease. This demonstrates that a hybrid design (PSDNorm in the first layers, BatchNorm in the rest) is empirically optimal, and that the remaining BatchNorm layers do not undermine PSDNorm’s effectiveness.
>
> Overall, applying PSDNorm to three layers provides the best balance between performance, variance reduction, and computational efficiency.
>
> This analysis offers additional insights into the behavior and effectiveness of the method, thanks for suggesting this.
>
> >• Channel independence: The method uses per-channel PSDs, from my understanding, as shown in Eq. 2. I think this assumes uncorrelated sensors. For learned feature maps, channel independence is unlikely to hold, so cross-channel correlations are ignored. I think this caps achievable alignment across channels.
>
> We emphasize that sleep staging relies on only two input channels, so spatial information is limited at the input level. Deeper in the network, the number of channels increases, but these channels represent mixtures of earlier features rather than true spatial dimensions. All other normalization layer treat channels and time as independently features, with PSDNorm we encode temporal correlation but treat channels independently.
>
> Moreover, introducing full spatio-temporal normalization would substantially increase computational cost. Our goal is to keep the normalization layer lightweight, which further motivates the design choice behind our method.
>
> >• Test-time adaptation: This point is more minor. The paper discusses TTA in Section 3.3. However, I think the comparison overlooks key issues with PSDNorm for TTA that make me uncertain about the claims in line 274. PSDNorm does not update its barycenter at inference (Algorithm 1), so it does not adapt to a novel target domain at test time.
>
> This is an important point and, in fact, a strength of our method. Traditional test-time adaptation approaches require additional training during inference, which can be impractical or even infeasible in many real-world settings.
>
> In contrast, our layer directly estimates the filter needed to project signals onto the reference PSD. The model then operates on a normalized PSD that is inherently robust to distribution shifts, all with minimal computational overhead and without any retraining needed.
>
>
> >Questions:
> • Hyperparameters: The paper fixes \alpha and f, with sensitivity analysis mainly on f. How important is \alpha? Other aspects like Welch window overlap or FFT length could be useful to discuss as well to get a sense of stability and cost.
>
> In our study we saw very small variation for different values of \alpha.
> For the Welch parameters, we used default parameters except the FFT length which is set to the filter size f.
>
>
> >• EEG channels: Will PSDNorm actually scale to other channels or modalities?
>
> We fully agree that incorporating additional modalities would be an interesting research direction. However, we have designed what we think are very advanced sleep staging experiments on SOTA models and with robust performance evaluation (Leave-One-Domain-Out) which correspond to substantial computational cost and require space to be presented properly. Adding an experiment on a new modality (images or videos), would require similar work including state-of-the-art baselines that would require substantial work and ressources that is not available to us.

---

> > ### Comment · Reviewer_yQGx · 2025-11-25
> >
> > I thank the authors for their careful response. Many aspects of the paper have been improved. Some concerns remain:
> >
> > •	PSDNorm Ablation: The authors justify not ablating PSDNorm completely as part of the "signal hierarchy". This explanation appears to be post-hoc. I agree with the fact that, in CNNs, early layers are closer to raw signals. And it's also true that aligning distributions near the input is often enough to get a lot of domain shift benefit. However, my read on your method is that it does not depend on being near the input. From what I can tell, Figure 10 might be the justification for using PSDNorm only in early layers. To me, this is a perfectly reasonable practical justification where PSDNorm just doesn't help, even if the comparison with TMA is slightly confounded. The authors still have not demonstrated that deep layers contain only abstract semantic features, whose domain shift is fully handled by BatchNorm. Moreover, my main complaint was about fairness/confounding comparisons with the baseline. After reading the response, I would say that this design choice is not obviously wrong, but I'm not sure that the current explanation gives a principled reason why PSDNorm should never be used throughout the network. My concerns here are largely resolved, though I would be curious about how and where PSDNorm should be applied in general.
> > •	Channel Independence: My concern here is not entirely resolved. The authors stress that there are only two raw channels, but this is only fair for input PSDs. Later feature maps can be highly correlated. Saying other norms also treat channels independently is only partially true (LayerNorm explicitly uses statistics across channels). I would also not say the assumption is harmless here. Furthermore, the paper makes fairly broad claims about the applicability of PSDNorm in general in the abstract and conclusion. But the only evidence is for a two-channel EEG. If the independence assumption really matters, then this is slightly overscoped.
> > •	TTA, Hyperparameters: Resolved, thank you.
> > •	Beyond EEG: Conceptually, this response is fine, but then I would prefer the paper to be tightened. Perhaps the abstract should be narrowed to "sleep EEG" or maybe "physiological time series".
> > I will raise my score to a 6 as the response mostly addresses my comments.

---

> > > ### Author Response · Authors · 2025-11-28
> > >
> > > Thank you very much for acknowledging our work and you positive feedback, and increasing your score.
> > >
> > > > PSDNorm placement
> > >
> > > We agree that the results in Figure 10 provide the strongest justification for applying PSDNorm in the early layers. We will revise the manuscript to present this primarily as an empirical design choice. We also agree with you that the interaction between PSDNorm and BatchNorm across layers is subtle, and characterizing optimal normalization placement constitutes an interesting research direction for the future. For now, and from empirical evidence, we advise to apply PSDNorm to the first layers.
> > >
> > >
> > > > Channel independence
> > >
> > > We acknowledge that assuming channel independence is a simplification. However, we note that PSDNorm shares this characteristic with standard layers like BatchNorm and InstanceNorm, which also normalize channels independently (as detailed in Appendix A.5). This is a standard design choice to maintain computational efficiency. While modeling cross-channel dependencies is indeed an interesting avenue for future research, it implies a different computational trade-off that lies beyond the scope of the efficient, temporal-focused PSDNorm proposed here.
> > >
> > > > Scope / Beyond EEG
> > >
> > > We will tone down the claims and be more clear in the abstract. We believe that PSDNorm can have an important impact outside of EEG but agree that we only show its interest on this data in the paper, which is actually reflected in the title ("...in Sleep Staging").

---

### Official Review · Reviewer_Q7SG · 2025-11-02

**Soundness:** 2
**Presentation:** 2
**Contribution:** 3
**Rating:** 4
**Confidence:** 4

**Summary:**

The paper proposes PSDNorm, a normalization layer for deep learning models on time-series signals, particularly EEG-based sleep staging. Traditional normalization methods (BatchNorm, LayerNorm, InstanceNorm) do not properly handle spectral distribution shifts arising from temporal autocorrelations and subject variability. PSDNorm addresses this issue by integrating Temporal Monge Alignment (TMA) within the network, allowing temporal context to be leveraged during normalization rather than as a separate preprocessing step. The method consists of three stages: 1) PSD estimation, 2) Running Riemannian barycenter update, and 3) Temporal Monge alignment. Experiments are conducted on 10 large-scale sleep datasets (10M samples, 10K subjects) under a Leave-One-Dataset-Out (LODO) setup. Experimental results show that PSDNorm outperforms all baseline methods, BatchNorm, LayerNorm, InstanceNorm, and TMA, demonstrating improved robustness under domain shifts.

**Strengths:**

- **S1:** The paper addresses an important problem, domain shift in physiological datasets, where variability across subjects and recording conditions poses a major challenge.
- **S2:** The paper is theoretically well-grounded, combining Monge mapping, Riemannian barycenters, and temporal alignment within a unified normalization framework. It is mathematically clear.
- **S3:** Consistent improvements on both USleep and CNNTransformer indicate its strong applicability to other architectures.
- **S4:** The analysis on subject-wise performance demonstrates the robustness of PSDNorm to inter-subject variability, one of the most difficult problems in physiological data modeling.
- **S5:** The idea of normalization based on spectral composition shows strong potential for sleep staging and other biomedical applications where temporal dependencies are critical.

**Weaknesses:**

- **W1:** The motivation behind PSDNorm is not entirely clear. While the paper states that PSDNorm is inspired by Temporal Monge Alignment (TMA), the concepts and roles of data preprocessing (TMA) and in-network normalization (PSDNorm) are fundamentally different. The rationale for integrating TMA into a normalization layer, and the specific benefits of this design are not sufficiently demonstrated.
- **W2:** Experiments are primarily conducted on large, well-curated datasets. Low-resource or few-subject scenarios are not adequately tested (the balanced@400 setting is still relatively large). For example, can the authors provide the performance of PSDNorm on balanced@40? While the authors argue that PSDNorm is effective in limited-data settings, the result in Table 2 does not fully support their argument (reports only balanced@400, which seems close to medium-scale rather than small-scale).
- **W3:** The normalization baselines are limited to BN, IN, and LN. Broader domain adaptation methods such as AdaBN, DSBN, or Deep CORAL would strengthen the comparative analysis.
- **W4:**  While PSDNorm targets dataset shift at test time, its behavior across datasets with highly divergent characteristics (Table 1) remains ambiguous. Performance trends in Table 2 do not clearly confirm improved robustness to severe distribution shifts.
- **W5:**  Although Figure 4 provides subject-level insight, since sleep datasets are often highly imbalanced, a class-wise performance analysis for poorly predicted subjects would provide more insight. The results from Table 2 and Table 5 indicate a higher F1-score compared to BAAC (balanced accuracy), which appears as certain classes dominating the overall improvements. Could the authors provide class-wise performance results for further clarification?

**Questions:**

Please address the above weaknesses.

---

> ### Author Response · Authors · 2025-11-22
>
> **W1**:
> The main difference between PSDNorm and TMA is that the normalization is done at different steps. Aligning the PSDs in the input space with TMA does not guarantee that the early network layers will not reintroduce a distribution shift. With PSDNorm, however, the shift induced in these first layers is substantially reduced since the normalization is done inside the network.
>
> To illustrate this effect, we added a visualization of the PSD at different stages of the network. The results show that using TMA alone is insufficient to address the shift that emerges inside the model. In contrast, normalization layers such as InstanceNorm or PSDNorm help realign the PSD throughout the network, highlighting the importance of normalizing beyond the input space.
>
> **W2**:
> We thank the reviewer for suggesting this experiment. We have added a study of the low-data regime in the Appendix. As shown in Table 14, PSDNorm with a filter size of 5 outperforms BatchNorm, and has similar performance as InstanceNorm. In this extreme setting, a larger filter size is needed to better mitigate the variability between subjects and avoid overfitting.
>
> This study further strengthens the understanding of our method.
>
> **W3**:
> Thank you for mentionning those potential baselines. We agree that adding AdaBN or DSBN would strengthen the analysis.
>
> In Table 15 of the Appendix (and in the next comment), we include a comparison with AdaBN using the authors’ official implementation. We adapted the BatchNorm statistics separately for each subject. In the original paper, all BN layers are replaced but for a more  AdaBN(3), which adapts only the first three BN layers.
>
> As expected, AdaBN struggles to achieve strong performance on sleep staging. It consistently underperforms compared to TMA and, in some cases, even performs worse than standard BatchNorm. Notably, increasing the number of adapted BN layers further degrades performance, highlighting the importance of not adapting too many layers within the model.
>
> We also want to point out that Deep Coral is actually not a direct competitor since it is a pairwise Domain Adaptation method that need additional training at inference, which is something that we are trying to avoid.
>
> **W4**:
>
> This is a very interesting question, but it introduces several complexities. First, it is difficult to define what constitutes the shift. Variability in the population between the datasets does not systematically produce a shift in the data that leads to a drop in performance.
>
> One example is MROS, which is composed only of older women but yields good performance even with BatchNorm. That can be explain that the sleep parameters remain largely unchanged over healthy older adults [1]. Conversely, for the CHAT dataset, composed only of children, the drop is significant, and we can observe that using PSDNorm helps considerably. This is consistent with known developmental changes in the PSD during adolescence [2]. Finding a method to link robustness to shift could be a major advance in the field, but we believe it is out of the scope of this paper.
>
> [1] Li et. al., Sleep in Normal Aging, 2017
>
> [2] Campbell et. al., Longitudinal trajectories of non-rapid eye movement delta and theta EEG as indicators of adolescent brain maturation, 2009
>
> **W5**:
>
> Thank you for raising this point. We have included class-wise performance tables for each method in the appendix, along with a barplot illustrating the relative difference in scores between our proposed normalization methods and BatchNorm.
>
> As expected, the N1 class (light sleep) remains the most challenging to classify. However, for all classes, PSDNorm consistently provides a greater improvement in the evaluation score compared to the other normalization methods tested.
> That's a very interesting observation that PSDNorm improves the performance on the easiest classes, like Wake, in addition to helping the hardest classes (like N1). This indeed strengthens your argument.

---

> > ### Author Response · Authors · 2025-11-24
> >
> > | Dataset   | BatchNorm           | LayerNorm         | InstanceNorm                | AdaBN(3)           | AdaBN(full)        | TMA                  | PSDNorm          |
> > |-----------|----------------------|-------------------|-----------------------------|--------------------|--------------------|-----------------------|-----------------------------|
> > | ABC       | 78.26±1.33           | 75.29±0.81        | **78.73±0.42**              | 78.25±1.30         | 76.89±1.30         | 78.04±0.51            | 78.18±0.68                  |
> > | CCSHS     | 87.42±0.16           | 85.20±0.48        | **87.62±0.42**              | 87.38±0.17         | 87.02±0.16         | 87.57±0.20            | 87.58±0.30                  |
> > | CFS       | 84.32±0.57           | 81.66±1.36        | **84.72±0.33**              | 84.21±0.60         | 84.05±0.56         | 84.58±0.20            | 84.29±0.36                  |
> > | CHAT      | 66.55±0.88           | 61.19±1.16        | 64.43±4.41                  | 66.49±0.89         | 66.44±0.88         | 68.73±2.48            | **70.28±1.70**              |
> > | HOMEPAP   | 75.25±0.50           | 74.86±0.25        | 76.47±0.63                  | 75.15±0.46         | 74.46±0.53         | 76.10±0.32            | **76.83±0.61**              |
> > | MASS      | 70.00±1.91           | 68.56±3.33        | 71.52±1.13                  | 69.68±1.66         | 68.31±1.86         | 71.63±1.92            | **72.77±1.09**              |
> > | MROS      | **80.37±0.20**       | 78.05±0.22        | 80.28±0.21                  | 80.34±0.20         | 80.27±0.20         | 80.09±0.40            | 80.26±0.11                  |
> > | PhysioNet | **75.81±0.13**       | 71.82±2.12        | 74.68±0.55                  | 75.27±0.14         | 74.01±0.14         | 75.31±1.54            | 74.82±2.11                  |
> > | SHHS      | 76.44±0.92           | 75.12±0.39        | 78.68±0.37                  | 76.43±0.92         | 76.41±0.92         | 77.00±0.39            | **78.88±0.68**              |
> > | SOF       | 81.08±1.14           | 78.70±0.50        | 80.68±1.38                  | 81.05±1.13         | 80.67±1.13         | **81.25±0.71**         | 79.49±0.41                  |
> > | Mean      | 77.22±0.34           | 75.04±0.42        | 78.17±0.28                  | 77.18±0.34         | 77.08±0.34         | 77.74±0.36            | **78.85±0.59**              |

---

### Meta-Review · Area_Chair_Aaco · 2025-12-12

**Summary:**

This paper proposes PSDNorm, a new normalization layer. It aligns the power spectral density of time-series to reduce distribution shifts. Reviewers appreciated the theoretical foundation, ablation study, and comparisons to standard normalization methods. They raised some concerns, including why spectral alignment is used as a layer and not just preprocessing. They also questioned why PSDNorm is only applied in early layers, its assumption of channel independence, and its comparison to other domain adaptation methods. Authors addressed these issues, in my opinion, effectely. They added new experiments in the low-data regime, compared PSDNorm to AdaBN,  provided visualizations of PSD across network layers, and performed sensitivity studies. The rebuttal greatly improved the paper's empirical support and clarified its design.

**Reviewer Concerns:**

- Authors tested PSDNorm with very little data and showed it still works well. A larger filter size helps when data is scarce.
- They compared to AdaBN, showcasing that PSDNorm still outperforms that method in most of the cases (Table 15).
- Ablation studies show that PSDNorm aligns PSD inside the network, unlike TMA or BatchNorm.
- Reviewers asked why PSDNorm is only in early layers. The authors made a new study (Figure 10) on the impact of the number of layers in U-Sleep using PSDNorm.

Some (relatively minor) issues remain.
- PSDNorm assumes signals are Gaussian and periodic, there not many comments regarding when and how this assumption may degrade performance.
- PSDNorm treats each channel separately, while this is a common feature across normalization layers. However, they do not study if correlations among channels can degrader performance.
- The paper only tests on EEG sleep data, while the method is presented as general for time-series models.
- I believe extension to SSMs (Mamba & Similar) would be an impact. Beyond of the scope of the paper though.

**Reviewer Scores:**

Reviewer Q7SG initially scored 4. After the rebuttal, which added low-data experiments and AdaBN comparisons. Likely raise the score to 6.

Reviewer yQGx started at 4. Authors clarified layer placement and channel independence assumptions. The reviewer explicitly raised the score to 6 in the text.

Reviewer vTBu gave an 8 initially and remained positive.

Reviewer ovsu scored 4 and was unsure about theoretical grounding and narrow evaluation. Not likely to change the score.

Reviewer 6ujs started at 6. After responses including AdaBN comparisons and expanded related work, the reviewer likely raise the score.

Average final score estimate: Approximately 6.2–6.4, above the acceptance threshold for ICLR.

---

### Decision · Program_Chairs · 2026-01-26

Accept (Poster)